# SMITE: Segment Me In Time

**Amirhossein Alimohammadi**[1]**, Sauradip Nag**[1]**, Saeid Asgari Taghanaki**[1,2]**,**
**Andrea Tagliasacchi**[1,3,4]**, Ghassan Hamarneh**[1]**, Ali Mahdavi Amiri**[1]
[1]Simon Fraser University [2]Autodesk Research [3]University of Toronto [4]Google DeepMind

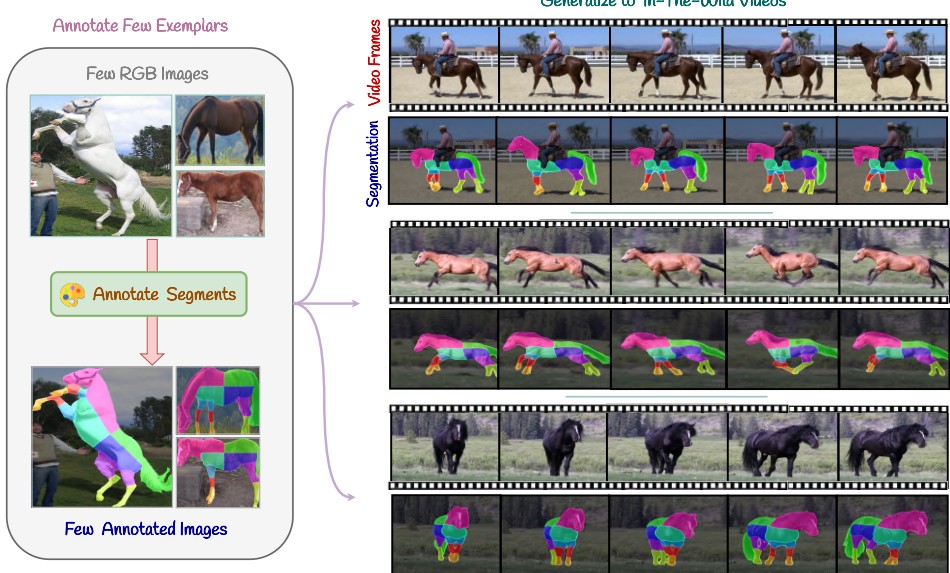

Figure 1: **SMITE.** Using only one or few segmentation references with fine granularity (left), our method learns to segment different unseen videos respecting the segmentation references.

## ABSTRACT

Segmenting an object in a video presents significant challenges. Each pixel must be accurately labeled, and these labels must remain consistent across frames. The difficulty increases when the segmentation is with arbitrary granularity, meaning the number of segments can vary arbitrarily, and masks are defined based on only one or a few sample images. In this paper, we address this issue by employing a pre-trained text to image diffusion model supplemented with an additional tracking mechanism. We demonstrate that our approach can effectively manage various segmentation scenarios and outperforms state-of-the-art alternatives. The project page is available at https://segment-me-in-time.github.io/.

## 1 INTRODUCTION

Segmenting an object in a video poses a significant challenge in computer vision and graphics, frequently employed in applications such as visual effects, surveillance, and autonomous driving. However, segmentation is inherently complex due to variations in a single object (scale, deformations, etc.), within the object class (shape, appearance), as well as imaging (lighting, viewpoint). In addition, difficulty arises due to the segmentation requirements, such as its granularity (i.e., number of segments), as demanded by the downstream tasks. For example, in face segmentation, one VFX application might need to isolate the forehead for wrinkle removal, while another, such as head tracking, might treat it as part of the whole face. Creating a comprehensive dataset for every possible segmentation scenario to develop a supervised segmentation technique is extremely time-consuming and labor-intensive. Therefore, there is a need to segment images or videos based on a reference image. We call this type of segmentation *flexible granularity*.

When flexible granularity segmentation is applied on a large scale, it significantly improves downstream tasks such as VFX production, which involves managing numerous shots and videos. By

segmenting one or a few reference images only once and then using those images to segment any video that features a target object of the same class, we can eliminate the need for separate segmentation of each video, thereby making the process far more efficient. In this paper, we tackle the challenge of video segmentation using one or few reference images that are not derived from the video frames themselves. For example, as shown in Fig. 1, a few annotated images are provided as references to our model, and our method, SMITE, successfully segments videos, exhibiting an object from the same class, in the same level of granularity. Importantly, none of the frames from these videos are included in the reference images, yet SMITE is capable of segmenting the videos with objects that exhibit different colors, poses, and even occlusions. This is an important feature when working with large-scale videos requiring consistent segmentation (such as VFX videos needing the same enhancements), since there is no need for manual intervention to segment each video's frames.

While recent work has explored flexible granularity segmentation for objects in images by leveraging the semantic knowledge of pretrained text-to-image diffusion models Khani et al. (2024), the complexity increases in videos. Ensuring label consistency across frames and managing instances where image segmentation may fail to produce accurate results require additional considerations.

To tackle these challenges and achieve consistent segmentation across frames, we utilize the semantic knowledge of pretrained text-to-image diffusion models, equipped with additional *temporal attentions* to promote temporal consistency. We also propose a *temporal voting* mechanism by tracking and projecting pixels over attention maps to maintain label consistency for each pixel. This approach results in segmentations with significantly reduced flickering and noise compared to per-frame segmentation methods while segments still follow the reference images thanks to our *low-pass regularization* technique that ensure preserving the structure of segments provided by attention maps and optimized according to the reference images.

Moreover, rather than simply optimizing a token for each segment Khani et al. (2024), we also *fine-tune cross-attentions* to enhance segmentation accuracy and better align with the reference images. Consequently, our method not only supports videos with temporal consistency but also outperforms flexible granularity image segmentation techniques in segmenting a single image.

We validate our design choices and methodology through comprehensive experiments detailed in the paper. As existing datasets with arbitrary semantic granularity are lacking, we introduce a small dataset, SMITE-50, to demonstrate the superior performance of our method against baselines. Additionally, we conduct user studies that highlight our method's effectiveness in terms of segmentation accuracy and temporal consistency.

## 2 RELATED WORK

**Part-based semantic segmentation.** In computer vision, semantic segmentation, wherein a class label is assigned to each pixel in an image, is an important task with several applications such as scene parsing, autonomous systems, medical imaging, image editing, environmental monitoring, and video analysis (Sohail et al., 2022; He et al., 2016; Chen et al., 2017a; Zhao et al., 2017; He et al., 2017; Chen et al., 2017b; Sandler et al., 2018; Chen et al., 2018; Ravi et al., 2024). A more fine-grained derivative of semantic segmentation is semantic part segmentation, which endeavors to delineate individual components of objects rather than segmenting the entirety of objects. Despite notable advancements in this domain (Li et al., 2023; 2022), a limitation of such methodologies is their reliance on manually curated information specific to the object whose parts they aim to segment. To solve the annotation problem, some works(Pan et al., 2023; Wei et al., 2024) proposed open-set part segmentation frameworks, achieving category-agnostic part segmentation by disregarding part category labels during training. Building on this, further works such as SAM (Kirillov et al., 2023), Grounding-SAM (Ren et al., 2024) explored utilizing foundation models to assist in open-vocabulary part segmentation. However, most of these methods can only segment the parts that are semantically described by text. With the influx of Stable Diffusion (SD) based generative segmentation approaches (Khani et al., 2024; Namekata et al., 2024), such issues have been partly solved by allowing SD features to segment semantic parts at any level of detail, even if they cannot be described by text. Despite such progress, applying such fine-grained segmentations on videos is challenging and unexplored. Our proposed SMITE presents the first part-segmentations in videos wherein it segments utilizing the part features from a pre-trained SD and generalizes it to any-in-the wild videos.

**Video segmentation.** Video segmentation methods can be categorized as video semantic segmentation (VSS) (Zhu et al., 2024; Zhang et al., 2023a;b; Li et al., 2024; Ke et al., 2023; Wang et al., 2024), video instance segmentation (VIS) (Yang et al., 2019) and video object segmentation (VOS) (Xie et al., 2021a; Wang et al., 2021b; Cheng et al., 2021c;d; Bekuzarov et al., 2023). VSS and VIS extends image segmentation to videos, assigning pixel labels across frames while maintaining temporal consistency despite object deformations and camera motion. VOS, in contrast, focuses on tracking and isolating specific objects throughout the video. Both tasks leverage temporal correlations through techniques like temporal attention (Mao et al., 2021a; Wang et al., 2021a), optical flow (Xie et al., 2021a; Zhu et al., 2017), and spatio-temporal memory (Wang et al., 2021b; Cheng & Schwing, 2022). Recent efforts, such as UniVS (Li et al., 2024), proposes unified models for various segmentation tasks, utilizing prior frame features as visual prompts. However, these methods struggle with fine-grained part segmentation and generalization to unseen datasets (Zhang et al., 2023b). Bekuzarov et al. (2023) leverage a spatio-temporal memory module and a frame selection mechanism to achieve high-quality video part segmentation with partial annotations. However, it requires frame annotations from the same video complicating video segmentation at scale. In contrast, we only need segmentation references for a few *arbitrary selected* images and we can segment an *unseen* given video respecting the segementation references. Therefore, per video manual annotation is not needed in our method.

**Video diffusion models.** Recently, diffusion models (Ho et al., 2020; Song et al., 2020a;b) have gained popularity due to their training stability and have been used in various text-to-image (T2I) methods (Ramesh et al., 2021; 2022; Saharia et al., 2022; Balaji et al., 2022), achieving impressive results. Video generation (Le Moing et al., 2021; Ge et al., 2022; Chen et al., 2023b; Cong et al., 2023; Yu et al., 2023; Luo et al., 2023) can be viewed as an extension of image generation with an additional temporal dimension. Recent video generation models (Singer et al., 2022; Zhou et al., 2022; Ge et al., 2023; Nag et al., 2023; Cong et al., 2024) attempt to extend successful text-to-image generation models into the spatio-temporal domain by inflating the T2I UNet. VDM (Ho et al., 2022) adopted this inflated UNet for denoising while LDM (Blattmann et al., 2023) implement video diffusion models in the latent space. Video diffusion models can be categorized into inversion-based and inversion-free methods. Inversion-based approaches (Cong et al., 2024; Jeong & Ye, 2023) use DDIM inversion to control attention features ensuring temporal consistency, while inversion-free methods (Zhang et al., 2023c) focus on more flexible conditioning, wider compatibility and better generation quality. However, inversion-free methods can suffer from flickering due to a lack of DDIM inversion guidance. Recent works (Wang et al., 2024; Zhu et al., 2024) explored inversion-free T2V diffusion models to segment objects in videos, but they fail to generate fine-grained segments and often produce flickering segmentation results. Building on this, our method solves the seminal problem of video part-segmentation which combines an inversion-free model coupled with point-tracking algorithms (Karaev et al., 2023) to generate consistent, generalizable, flicker-free segmentations.

# 3 PRELIMINARIES

**Latent diffusion models and WAS maps.** Latent Diffusion Models perform the denoising operation on the latent space of a pretrained image autoencoder. Each latent pixel corresponds to a patch in the generated image. Starting from pure random noise $z_T$, at each timestep $t$, the current noisy latent $z_t$ is passed through a denoising UNet ($\epsilon_\theta$), which is trained to predict the current noise $\epsilon_\theta(z_t, y, t)$ using text prompt $y$. In each block, the UNet employs residual convolution layers to generate intermediate features, which are then fed into attention layers. These attention layers average various values based on pixel-specific weights. The cross-attention layers (denoted by $A_{ca}$) incorporate semantic contexts from the prompt encoding, whereas the self-attention layers (denoted by $A_{sa}$) leverage global information from the latent representation itself. SLiMe (Khani et al., 2024) demonstrated that text embeddings can be learned from a few images to segment other unseen images by leveraging both cross attention and self-attention layers. In fact, each segment of an image is linked with a text token that is optimized to align its associated cross-attention and self-attention maps with the corresponding segment in the provided training image. For segmentations with multiple levels of granularity, multiple tokens are optimized, with each token representing a distinct segment. The optimization process of attention maps is captured in a novel representation called Weighted Accumulated Self-Attention (WAS) map, $S_{\text{WAS}}$. This is defined as follows:

$$S_{\text{WAS}} = \text{Sum}(\text{Flatten}(R_{ca}) \odot A_{sa}), \tag{1}$$

where $R_{ca}$ is the downsampled latent of $A_{ca}$.

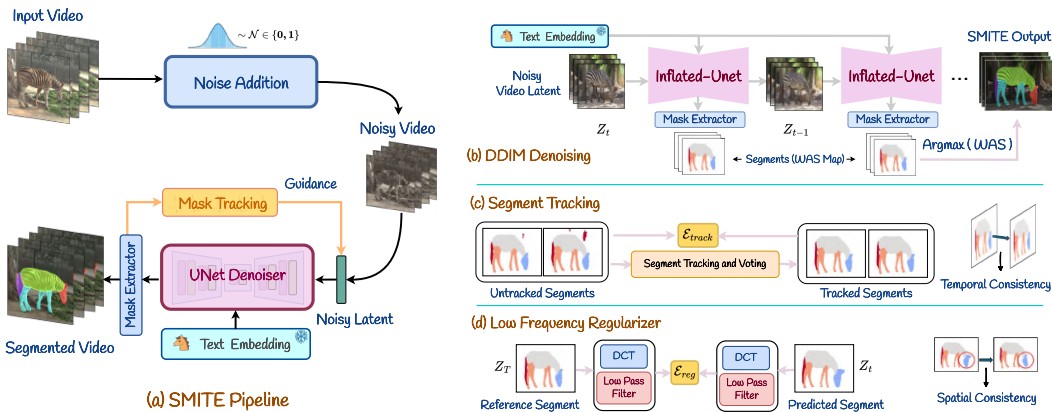

Figure 2: **SMITE pipeline.** During inference (a), we invert a given video into a noisy latent by iteratively adding noise. We then use an inflated U-Net denoiser (b) along with the trained text embedding as input to denoise the segments. A tracking module ensures that the generated segments are spatially and temporally consistent via spatio-temporal guidance. The video latent $z_t$ is updated by a tracking energy $\mathcal{E}_{track}$ (c) that makes the segments temporally consistent and also a low-frequency regularizer (d) $\mathcal{E}_{reg}$ which guides the model towards better spatial consistency.

**Inflated UNet.** A T2I diffusion model, such as LDM (Rombach et al., 2022), usually utilizes a U-Net(Ronneberger et al., 2015) architecture, which involves a downsampling phase, followed by an upsampling with skip connections. The architecture consists of layered 2D convolutional residual blocks, spatial attention blocks, and cross-attention blocks that incorporate textual prompt embeddings. To extend the T2I model for T2V tasks, the convolutional residual blocks and spatial attention blocks are inflated. Following earlier approaches (Cong et al., 2024; Wu et al., 2022), the $3 \times 3$ convolution kernels in the residual blocks are adjusted to $1 \times 3 \times 3$ by introducing a pseudo temporal channel. To enhance the temporal coherence, we further extend the spatial self-attention mechanism to the spatio-temporal domain. The original spatial self-attention method focused on patches within a single frame. However, in inflated UNet, we use all patch embeddings from the entire video as the queries, keys, and values. This allows for a full understanding of the video's context. Additionally, we reuse the parameters from the original spatial attention blocks in the new dense spatio-temporal attention blocks.

## 4 METHOD

Here, we first introduce and formalize our method (SMITE) designed for achieving temporally consistent video segmentation with varying levels of granularity, guided by one or more reference images (Sec. 4.1). To achieve this and capture fine-grained segments, we first propose a new training strategy applied on the inflated UNet (Sec. 4.2). The segmentation obtained from the inflated UNet may lack temporal consistency. To address this, we employ a voting mechanism guided by a tracking method that is projected onto the attention maps. However, relying solely on tracking to adjust the segments might lead to deviations from the provided samples. To mitigate this, we incorporate a frequency-based regularization technique to maintain detailed segmentations across frames (Sec. 4.3). Since tracking and frequency-based regularization may pull the segmentation in different directions, we use a energy based guidance optimization technique to balance both approaches (Sec. 4.4).

### 4.1 PROBLEM SETTING

**Problem statement.** Given one or few images, $\mathcal{I}_n \in \mathbb{R}^{H \times W \times 3}$, of a subject, along with its segment annotations $\mathcal{Y}_n = \{\mathcal{Y}_n^i | i = 1 : K\}$ where $\mathcal{Y}^i$ denotes binary segment mask $i$, and $n$ and $K$ are respectively the number of images and segments. Our objective is to learn temporally consistent segments of the subject for a given video $\mathcal{V} = \{v_{j=1}^M\}$ where $v_j$ represents video frames.

**Our framework.** We use Stable Diffusion's (SD) semantic knowledge to learn the segments defined by few images and then generalize them to the video of the subject in any pose, color or size. This

implies that the model needs to share information across frames to enforce temporal consistency. Differently from the UNet structure used in Khani et al. (2024), we apply an inflation to the T2I models across the temporal dimension (Wu et al., 2022) to enable temporal attention across all video frames. Also, we incorporate a tracking module combined with low-frequency regularization to enhance spatio-temporal consistency across frames. The overall inference pipeline of our model, SMITE, is illustrated in Fig 2.

## 4.2 LEARNING GENERALIZABLE SEGMENTS

We first learn segments provided by the reference images of the subject by optimizing a text embedding for each segment of the reference images that can be used for segmenting the given videos. We also fine tune the cross attentions of the SD to better match the provided segments since the text embeddings alone may not be able to fully capture the masks' details.

**Learning text embeddings.** We begin by passing the reference images $\mathcal{I}$ and text embeddings $\mathcal{T}$ into into SMITE (denoted by $\psi(.)$) . Similar to SLiME (Khani et al., 2024), we obtain image resolution WAS maps (denoted by $\mathcal{S}$) from our inflated UNet as follows:

$$\mathcal{S} = \psi_\theta(\mathcal{I}, \mathcal{T}) \quad , \quad \hat{Y} = argmax(\mathcal{S}), \tag{2}$$

where $\theta$ represents the learnable parameters of the model and $\mathcal{S}$ is defined the same as Eq. 1. Text embeddings $\mathcal{T}$, which are initialized randomly or with the names of segments, correspond to segment masks $\mathcal{S} = [S_{\text{WAS}}^1, S_{\text{WAS}}^2, ..., S_{\text{WAS}}^K]$, and $\hat{Y}$ is the segmentation output. Since the inflated UNet is designed for videos, we pass reference images in $\mathcal{I}$ to $\psi(.)$ as videos with a single frame. Together with the ground-truth mask $\mathcal{Y}$, we find the optimized text embeddings, $\mathcal{T}^*$, that are correlated with the segments in $\mathcal{S}$.

**Network fine-tuning.** Only learning text embeddings fails to capture complex granularities. To extract more customizable fine-grained segments from the video, we need to fine-tune the existing SD weights (denoted by $\theta$) using the available provided segment annotations. As shown in the inset figure, optimizing solely the text embeddings struggles with asymmetrical segmentation (e.g., segmenting only one eye). We hypothesize that such issues arises

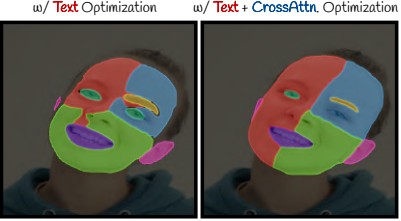

because the model may get stuck in local minima when relying exclusively on text embeddings for optimization. To mitigate this, we update the cross-attention layers $A_{ca}$ in the UNet along with the text-embedding but do so in two phases. First, the model is frozen, while the text embeddings corresponding to the segments denoted by $\mathcal{T}$ are optimized using a high learning rate. Thus, an initial embedding ($\mathcal{T}^*$) is achieved quickly without detracting from the generality of the model, which then serves as a good starting point for the next phase. Second, we unfreeze the cross-attention weights and optimize them along with the text embeddings, using a significantly lower learning rate. This gentle fine-tuning of the cross-attention and the embeddings enables faithful generation of segmentation masks with varied granularity. For both training phases, we use the same combination of losses ($\mathcal{L}_{\text{CE}}$, $\mathcal{L}_{\text{MSE}}$ and $\mathcal{L}_{\text{LDM}}$) used in SLiME (Khani et al., 2024). This results in an optimized SMITE model (denoted by $\Psi_\theta^*$) generating generalizable segment across different videos.

## 4.3 TEMPORAL CONSISTENCY

To enhance temporal consistency, we use temporal attention in SMITE's Inflated UNet denoiser. While it improves temporal consistency compared to independent frame processing (Fig 3(a)), inconsistencies remain due to the need for segmenting at different granularities, often with imprecise boundaries. These inconsistencies can cause flickering or unnatural transitions (Fig 3(b)).

**Segment tracking and voting.** The first step to ensure segment consistency is tracking the segments across time. Point tracking methods like CoTracker (Karaev et al., 2023) are well-suited for our approach because they use point correspondences to minimize pixel drift over time. However, since our segments come from attention maps, tracking needs to occur directly on these maps. Since CoTracker is trained on spatial domains, we first apply CoTracker (denoted by $\mathcal{P}$) on the frames of video $\mathcal{V}$ bidirectionally as $\{X, Y\} = \mathcal{P}(\mathcal{V})$ where $X, Y$ are set of tracked pixel coordinates. To project these tracking onto the the attention maps, we use a sclaing operator $\phi(.)$ and linearly scale the

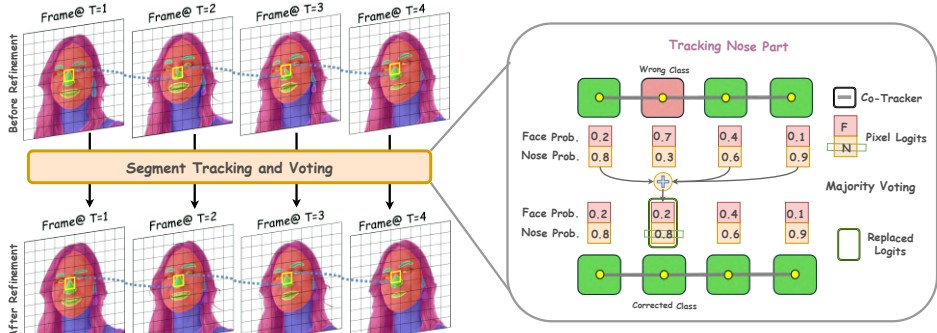

Figure 4: **Segment tracking module** ensures that segments are consistent across time. It uses co-tracker to track each point of the object's segment (here it is nose) and then finds point correspondence of this segment (denoted by blue dots) across timesteps. When the tracked point is of a different class (e.g,. face) then it is recovered by using temporal voting. The misclassified pixel is then replaced by the average of the neighbouring pixels of adjacent frames. This results are temporally consistent segments without visible flickers.

pixel trajectories onto the attention space. We process the entire video in a temporal sliding window fashion having window length $w$. For each of such temporal window, we then leverage the projected trajectories and update the attention pixel's label based on the most frequent label it receives across the visible frames in the window. Formally, for a pixel with coordinates $(x_t, y_t)$ on the attention map latent at timestep $t$, its coordinates on all subsequent latents in the window can be derived from the tracker. The coordinates are linked, and the trajectory sequence can be presented as:

$$\{(x_{t-\frac{w}{2}}, y_{t-\frac{w}{2}}), ..., (x_t, y_t), ..., (x_{t+\frac{w}{2}}, y_{t+\frac{w}{2}})\} = \phi(\mathcal{P}(\mathcal{V})), \tag{3}$$

We choose the point corresponding to the center frame of the window $(x_t, y_t)$ as the first point and leverage bidirectional correspondences within the window. Each of these trajectory points $(x, y)$ is assigned a segment label $\hat{Y}$. However, certain pixels in the window may disappear and reappear over time as shown in Fig 4. To handle such scenarios, we discard labels for which the tracking pixel is invisible and then use *temporal voting* to update its segment labels. Therefore, we update the WAS maps (denoted by $\mathcal{S}_{\text{WAS}}$) corresponding to the tracking pixel $(x_t, y_t)$ and use the following formula to assign the labels:

$$\mathcal{S}_{\text{Tracked}}(x_t, y_t) = \text{Avg}(\mathcal{S}(x_l, y_l) \mid \hat{Y}(x_l, y_l) = F) \quad \forall l \in (t - \frac{w}{2}, t + \frac{w}{2}), \tag{4}$$

where $F$ denotes the most frequent label in the window.

After correcting the unstable pixel tracking over multiple-step denoising, we obtain a consistent set of segments ($\mathcal{S}_{tracked}$). Note that the window size $w$ is small (e.g, seven), hence to track the segments for the entire video, we need to slide the windows and track in an iterative fashion. More details are provided in the supplementary material.

Intuitively, if we would have relied only on cross-attention for segmentation, we could have adjusted it directly to align with the tracking output. However, since segments $\mathcal{S}$ are derived from the combination of cross and self-attention, direct manipulation is challenging. To tackle this, we propose an energy function to measure how well the segments before tracking (denoted by $\mathcal{S}$) align with the segments after correcting (denoted by $\mathcal{S}_{tracked}$) it using temporal voting. This energy is defined as:

Figure 3: **Video** best viewed in *Acrobat*.

$$\mathcal{E}_{\text{Tracking}} = CE(\text{S}, \text{S}_{\text{tracked}}), \tag{5}$$

where $CE(.)$ represents cross-entropy objective. This optimization ensures that the segment drift is corrected throughout the video to make it temporally consistent with low flicker.

**Low-pass regularization.** When tracking is applied, label modifications should be repeated through several denoising steps to ensure full temporal label propagation and achieve better consistency.

This process may cause deviations (Fig 3) from the original segmentation provided by the reference images that is captured in the segments ($\mathcal{S}$) in the initial denoising step. Our goal is to preserve the overall structure of the segments $\mathcal{S}$ from the WAS maps while smoothing boundary transitions for temporal consistency via tracking (refer to Fig 8). Low Pass Filters (LPF) have been effective in video generation to maintain temporal correlations and spatial frame structure (Wu et al., 2023; Si et al., 2024). Therefore, we use an LPF on the finally denoised segments $\mathcal{S}$ ensuring the structure of the LPF on the segments $\mathcal{S}$ predicted in the initial denoising step (denoted by $\mathcal{S}_{ref}$) is respected by the following function:

$$\mathcal{E}_{\text{Reg}} = \|\omega(\mathcal{S}) - \omega(\mathcal{S}_{ref})\|_1 \tag{6}$$

where $\omega(S) = \mathcal{H}_l \odot \text{DCT}_{3D}(\text{S})$, $\text{DCT}_{3D}$ refers to Discrete Cosine Transform and $\mathcal{H}_l$ refers to an LPF. This ensures that the segments structure is progressively corrected across the video frames at the end of the denoising phase as illustrated in Fig 3.

## 4.4 SPATIO-TEMPORAL GUIDANCE

Our overall energy function is minimized at test time by backpropagating through the diffusion process, as described in Chen et al. (2023a); Safaee et al. (2023), which updates the latent representation to achieve more consistent segmentation over time:

$$\mathcal{E}_{\text{Total}} = \lambda_{\text{Tracking}} \cdot \mathcal{E}_{\text{Tracking}} + \lambda_{\text{Reg}} \cdot \mathcal{E}_{\text{Reg}}, \tag{7}$$

where $\lambda_{\text{Tracking}}$ and $\lambda_{\text{Reg}}$ are the coefficients for the tracking and reference loss functions, respectively. We have two different energy functions: $\mathcal{E}_{\text{Tracking}}$ which reduces flicker and ensures segment consistency over time, while $\mathcal{E}_{\text{Reg}}$ which maintains correct semantics while preventing segment drift. Thus, optimizing $\mathcal{E}_{\text{Total}}$ encourages better spatial and temporal consistency in the WAS attention maps (S). Specifically, we update the latent $z_t$ using gradient descent on $\mathcal{E}_{\text{Total}}$:

$$z'_t \leftarrow z_t - \alpha_t \cdot \nabla_{z_t} \mathcal{E}_{Total}, \tag{8}$$

where $\alpha_t$ is the learning rate. Results indicate that this strategy leads to superior performance compared to when we either omit the voting mechanism or do not apply low-pass regularization.

## 5 RESULTS AND EXPERIMENTS

**Dataset and benchmark.** To evaluate our method, we introduce a benchmark dataset called SMITE-50, primarily sourced from Pexels. SMITE-50 features multi-granularity annotations and includes visually challenging scenarios such as pose changes and occlusions. To our knowledge, no existing datasets focus exclusively on multi-granular and multi-segment annotations. While the PumaVOS dataset Bekuzarov et al. (2023) contains a limited number of multi-part annotated videos across a diverse range, it does not offer multiple videos for specific granularities and categories.

We focus on three main categories: (a) Horses, (b) Human Faces, and (c) Cars, encompassing **41** videos. Each subset includes ten segmented reference images for training and densely annotated videos for testing. The granularity varies from human eyes to animal heads, etc. relevant for various applications such as VFX (see Fig. 5). All segments are labeled consistently with the part names used in existing datasets. Additionally, we provide *nine* challenging videos featuring faces with segments that cannot be described textually, as shown in Fig. 3 (Non-Text). Overall, our dataset comprises **50** video clips, each at least five seconds long. For dense annotations, we followed a similar approach to (Ding et al., 2023; Bekuzarov et al., 2023), creating masks for every fifth frame with an average of six

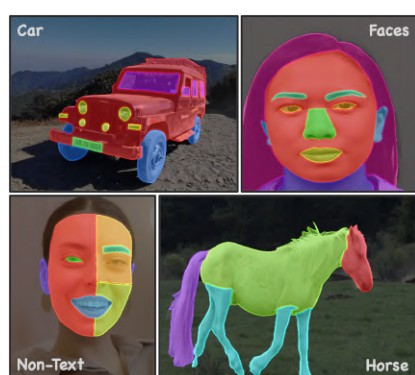

Figure 5: *SMITE-50* Dataset sample.

parts per frame across three granularity types (more info in Appendix). While PumaVOS dataset Bekuzarov et al. (2023) has 8% annotations, our SMITE-50 dataset has 20% dense annotations.

**Evaluation protocol.** In our setting, few reference images per class are used to train SMITE and then it is evaluated on videos from the same categories but not the same objects in the training data.

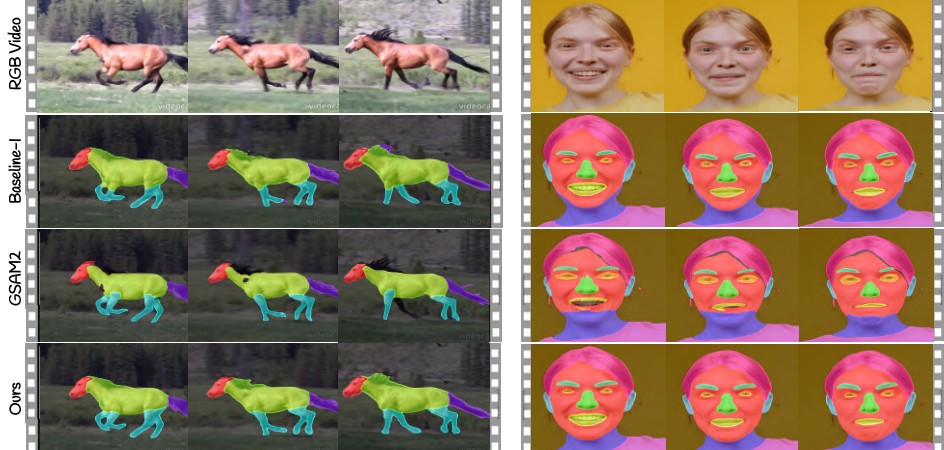

Figure 6: Visual comparisons with other methods demonstrate that SMITE maintains better motion consistency of segments and delivers cleaner, more accurate segmentations. Both GSAM2 and Baseline-I struggle to accurately capture the horse's mane, and GSAM2 misses one leg (Left), whereas our method yields more precise results. Additionally, both alternative techniques create artifacts around the chin (Right), while SMITE produces a cleaner segmentation.

For all the cases, we report the standard metrics (higher is better): Mean Intersection Over Union (mIOU), and Contour Accuracy $F_{measure}$ respectively.

**Quantitative comparison.** Since there are no available few-shot video part segmentation methods that can be applied to our setting, we build the multiple baselines for quantitative comparison. First, a few-shot image segmentation based approach, in which **SLiMe** (Khani et al., 2024) is applied to the video frame-by-frame. We term this as *Baseline-I*. Second, **Grounded SAM2** (Ren et al., 2024; Ravi et al., 2024), a recently introduced zero-shot foundation model for video segmentation approach which is applied directly on the video. For *Baseline-I*, we applied the same training procedure and used *SMITE-50* dataset. Another baseline, *Baseline-II* is to use SLiMe and CoTracker simultaneously. However, directly applying tracking in the pixel space does not produce improved results, as it can lead to drift in segmentation. In contrast, SMITE leverages tracking on the attention maps, focusing on where the object and its parts are located, which minimizes the impact of pixel color shifts. Additionally, we have incorporated a regularization mechanism into our tracking process to further mitigate such drifts, ensuring alignment with the true semantic segmentations. We have also provided another baseline, *Baseline-III*, which is the modification of SLiMe by only changing its UNet to the Inflated UNet. As shown in Tab. 1, our method produces the most accurate results compared to other baselines across all categories and metrics.

Generally, XMem++ (Bekuzarov et al., 2023) cannot be utilized in our setting since we do not want to use any frames from the provided video. However, in the Appendix, we included a comparison with XMem++ on a subset of their proposed dataset PUMaVOS with flexible granularity to provide a contrast and show that our method is effective on datasets beyond SMITE-50. Although XMem++ is a semi-supervised technique, our method performs comparably and even outperforms it with fewer frames (e.g., a single shot). When XMem++ is given more frames per video (e.g., 10), its performance slightly surpasses ours, which is expected since SMITE does not require any substantial video pre-training. However, with a smaller number of frames (e.g., one or five), our method either outperforms or matches the performance of XMem++. Additionally, we report our performance on the image segmentation task in the Appendix, showing that we outperform SLiME (Baseline-I), highlighting the effectiveness of our design choices, including the cross attention optimization.

**Qualitative comparisons.** Qualitative comparisons are presented in Fig. 6. GSAM2 struggles to locate the boundaries accurately and produces coarse segments. SLiMe can usually segment the first frame accurately but struggles to preserve temporal consistency. Our method produces the best segmentation maps in terms of segmentation quality and temporal consistency. The maps have sharper boundaries and clean clusters compared to other methods. Our SMITE performs better than its SLiMe counterpart. In Fig. 7, we present results from other categories, demonstrating the versatility of

Table 1: **Quantitative evaluation on SMITE-50 dataset.** The results are presented for each category (Face, Horse, Car, Non-Text) having 10 reference image during training.

| Methods | Faces | | Horses | | Cars | | Non-Text | | Mean | |
|---|---|---|---|---|---|---|---|---|---|---|
| | $F_{meas.}$ | mIOU | $F_{meas.}$ | mIOU | $F_{meas.}$ | mIOU | $F_{meas.}$ | mIOU | $F_{meas.}$ | mIOU |
| Baseline-I | 0.81 | 72.95 | 0.64 | 65.48 | 0.57 | 61.38 | 0.67 | 66.69 | 0.67 | 66.62 |
| Baseline-II | 0.78 | 71.08 | 0.62 | 64.01 | 0.65 | 61.95 | 0.65 | 64.96 | 0.67 | 65.50 |
| Baseline-III | 0.82 | 73.25 | 0.64 | 67.48 | 0.70 | 68.49 | 0.66 | 68.22 | 0.70 | 69.36 |
| GSAM2 | 0.73 | 63.28 | 0.76 | 72.76 | 0.64 | 63.56 | - | - | 0.71 | 66.53 |
| **Ours** | **0.89** | **77.28** | **0.79** | **75.09** | **0.82** | **75.10** | **0.77** | **73.08** | **0.82** | **75.14** |

Table 2: **Network Ablation on SMITE-50 dataset.** Ablations on the entire SMITE-50 dataset. We respectively show the impact of inflated UNet (IUNet), cross attention tuning (CA Tuning), Tracking, and Low-pass (LP) regularization.

| Method | Baseline I (SLiMe) | +IUNet (Baseline III) | +CA Tuning | +Tracking | +LP Reg (SMITE) |
|---|---|---|---|---|---|
| **mIOU** | 66.22 | 69.36 | 71.70 | 72.64 | **75.14** |

our model in handling various object categories. As shown, despite significant differences in pose, expression, gender, and other properties between the video and the annotated images, our method still delivers high-quality results. This is particularly evident in the pineapple example, where the object is cut in half, yet the method successfully tracks it and produces accurate segmentation. More qualitative results are provided in Appendix 7.4.

**Ablation study.** To demonstrate the effectiveness of our design choices, we conducted a comprehensive ablation study showing the impact of each component of our method one by one on the entire SMITE-50 dataset. Tab. 2 shows that our final setting provides the highest accuracy while the impact of each component is noticeable. Note that the low-frequency regularization (LP Reg) helps maintain spatial structure and prevents segment drift when tracking is applied to $S_{WAS}$. Without tracking, LP Reg has minimal effect since no segment drift occurs in WAS maps (thus change of energy in $\mathcal{E}_{Reg}$ is almost zero), so we do not ablate LP Reg separately without tracking. We have also performed an ablation on the number of training images illustrated in Tab. 3. As expected the accuracy improves when the number of training samples increases.

**User study.** We conduct a user study because human judgment is best for assessing perception when both temporal consistency and faithfulness to segmentations are important. We collected 16 segmented videos (4 from Non-Text category) and asked 25 participants to rank the methods (i.e., 1 best and 5 worst) based on *segmentation quality* (i.e., fidelity to the segmentation reference) and *motion consistency* (i.e., reduced flicker), encouraging them to prioritize segmentation quality in their evaluations in terms of a tie in motion consistency. We do not report the scores of GSAM2 on Non-Text segments as the segments in the videos are not describable by texts. Tab. 4 demonstrates that SMITE achieves higher preference both for textual and non-textual segmentation.

Table 3: **Few shot ablation on SMITE-50.** The performance increases with more training images but still performs well in one shot setting.

| Training sample # | mIOU |
|---|---|
| 1-shot | 58.55 |
| 5-shot | 73.88 |
| **10-shot** | **75.14** |

Table 4: **User study.** We are ranked the best for both textual and Non-Text classes.

| Methods | Motion Consistency | |
|---|---|---|
| | Horse, Car, Face | Non-Text |
| Baseline-I | 3.59 | 2.41 |
| Baseline-II | 3.42 | 3.24 |
| Baseline-III | 2.98 | 3.32 |
| GSAM2 | 3.37 | - |
| **Ours** | **1.46** | **1.03** |

# 6   CONCLUSION, LIMITATIONS, FUTURE WORK

In this work, we introduce SMITE, a video segmentation technique that supports flexible granularity segmentation of objects within a video. SMITE leverages the semantic knowledge of a pre-trained

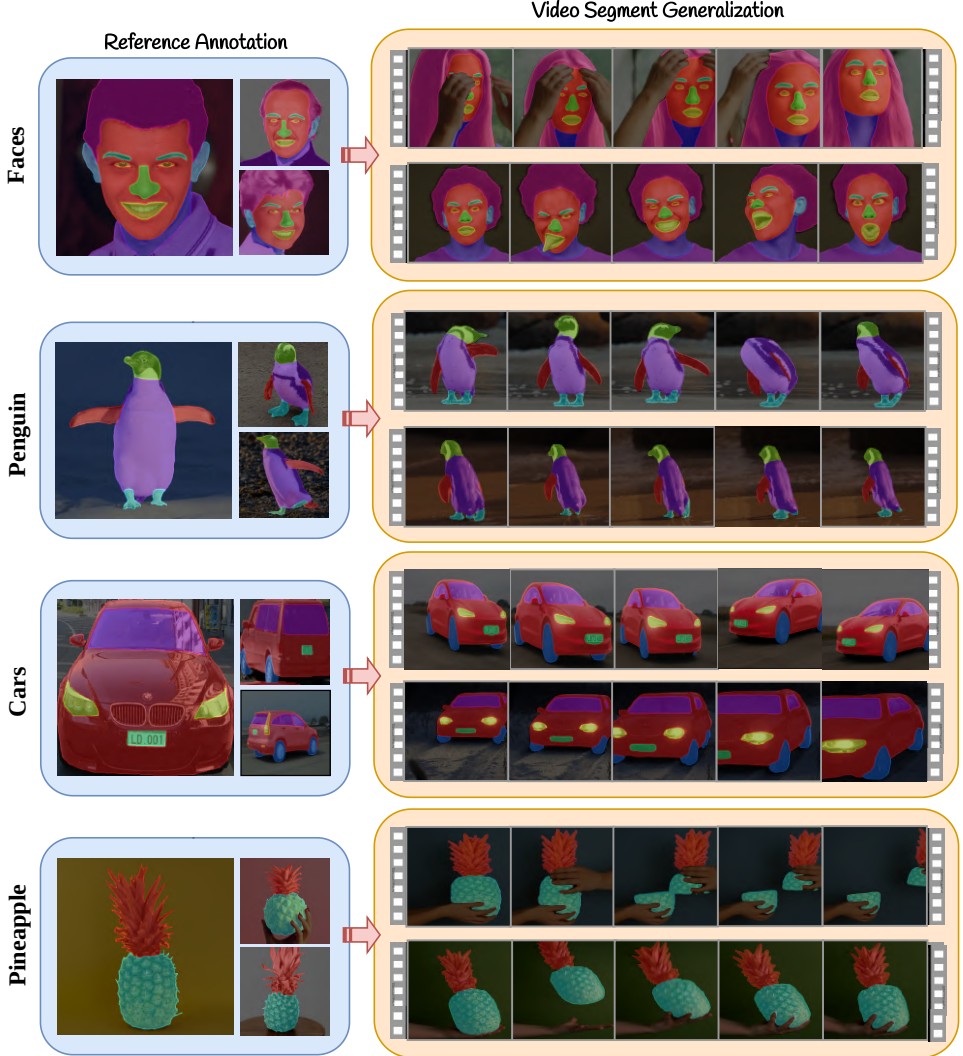

Figure 7: **Additional results.** We visualize the generalization capability of SMITE model (trained on the reference images) in various challenging poses, shape, and even in cut-shapes.

text-to-image diffusion model to segment a video with minimal additional training (i.e., only a few images), while utilizing the temporal consistency of the video to maintain motion consistency within the segments. Notably, SMITE can be trained with a few images that are not necessarily from the video, yet it effectively segments objects within an unseen video. To better show the capabilities of our method, we also collected a flexible granularity dataset called SMITE-50 that will be publicly available along with our code. Through various quantitative and qualitative experiments, as well as user studies, we demonstrate the effectiveness of our method and its components. However, our method still faces limitations that motivate future work. Firstly, although, our regularization ensures that tracking does not dominate the segmentation process or deviate from the anticipated semantic segmentations, since SMITE utilizes an off-the-shelf tracking technique, its tracking mechanism and voting system are still limited by the performance of the underlying tracking system to some extent. Secondly, the generation of WAS maps involves the use of low-resolution cross-attention mechanisms, which may result in reduced accuracy when segmenting fine details or small segments, such as tiny strings or minor components. Thirdly, it cannot offer an interactive segmentation since it involves the diffusion and optimization processes. In this work, we did not particularly focus on optimizing training and inference time but this can be an interesting future work.

**Acknowledgements.** This work was supported by the Natural Sciences and Engineering Research Council of Canada (NSERC) Discovery Grant. We sincerely thank Pouria Laghaee for his invaluable contributions and assistance in dataset annotation. We also appreciate the support of Helena Dehkharghanian, Ali Mohammadzade Shabestari, Soniya Almasi, and Fatemeh Kamani in the annotation process.

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

# 7 APPENDIX

## 7.1 SMITE-50 DATASET

*SMITE-50* is a video dataset that covers challenging segmentation with multiple parts of the object is being segmented in difficult scenarios like occlusion. It consists of 50 videos, up to 20 seconds long, from 24 frames to 400 frames with different aspect ratios (both vertical and horizontal). Frame samples of the dataset are provided in Fig5 in the main paper. Primarily our dataset consists of 4 different classes "Horses", "Faces", "Cars" and "Non-Text". Out of these sequences, "Horses" and "Cars" have videos which have been captured outdoors hence it has challenging scenarios including occlusion, view-point changes and fast-moving objects with dynamic background scenes. "Faces" sequence on the other hand have videos with occlusion, scale changes and more fine-grained parts which are difficult to track and segment across time. The "Non-Text" category has videos which have parts that cannot be described by natural language cues. Hence, these videos are difficult for zero-shot video segmentation (Ren et al., 2024) approaches as most of the models are reliant on textual vocabularies for the segmentations. As compared to existing VOS datasets (refer to Table 5 like DAVIS-2016 Perazzi et al. (2016), PUMAVOS Bekuzarov et al. (2023) and Youtube-VOS Xu et al. (2018a), SMITE-50 has more parts per video and many multi-part videos. Although Youtube-VOS (YT-VOS) has more videos but all of the videos lack multi-part annotations. SMITE-50 is a working progress and we intend to expand it beyond what it currently includes and make it publicly available.

Table 5: Comparison of various VOS datasets with SMITE-50.

| Dataset | DAVIS-2016 | PUMAVOS | YT-VOS | SMITE-50 |
|---|---|---|---|---|
| # training/testing videos | 30/20 | 0/24 | 3945/508 | **0/50** |
| Avg parts per video | 1 | 1.08 | 1 | **6.39** |
| Number of multi-part videos | 0 | 1 | 0 | **50** |

## 7.2 EXTRA ABLATIONS

**Performance on PUMaVOS and comparison with XMem++.** Here, we provide a comparison with XMem++ on their proposed dataset PUMaVOS to show that our method is effective on other datasets than our proposed SMITE-50. For this evaluation, we considered the following seven video splits namely :*Chair*, *Full face 1*, *Full face 2*, *Half face 1*, *Half face 2*, *Long Scene scale* and *Vlog* repetively. We chose the following categories as the object parts in these videos are not too small and also have flexible granularity. Note that XMem++ is a semi-supervised video segmentation technique meaning that it has been trained on largescale video segmentation datasets such as DAVIS (Pont-Tuset et al., 2017) or YoutubeVOS (Xu et al., 2018b). Nevertheless, SMITE performs comparably and even outperforms it in case of working with fewer frames (e.g., a single shot; see Tab. 6). Predictably, when XMem++ is provided with more frames per video (e.g., 10), its performance slightly exceeds ours since SMITE is not trained on a large segmentation dataset. However, with a smaller number of frames (e.g., one or five), our method either outperforms or matches the performance of XMem++. See Tables 6 and 7, for the quantitative comparisons.

Table 6: Comparison on a subset of PUMaVOS dataset (Bekuzarov et al. (2023)) when only one frame is used for training.

| Method | Chair $F_{meas.}$ | mIOU | Full face 1 $F_{meas.}$ | mIOU | Full Face 2 $F_{meas.}$ | mIOU | Half Face 1 $F_{meas.}$ | mIOU |
|---|---|---|---|---|---|---|---|---|
| GSAM2 | 0.49 | 58.82 | 0.99 | 97.47 | 0.94 | 94.78 | 0.29 | 57.66 |
| Baseline-I | 0.46 | 73.15 | 0.61 | 85.23 | 0.7 | 86.9 | 0.02 | 82.83 |
| XMem++ | 0.99 | 95.72 | 0.71 | 90.75 | 0.80 | 89.92 | 0.82 | 90.52 |
| Ours | 0.32 | 63.32 | 0.98 | 96.46 | 0.85 | 90.38 | 0.55 | 79.75 |

| Method | Half Face 2 $F_{meas.}$ | mIOU | Long Scene Scale $F_{meas.}$ | mIOU | Vlog $F_{meas.}$ | mIOU | Mean $F_{meas.}$ | mIOU |
|---|---|---|---|---|---|---|---|---|
| GSAM2 | 0.54 | 74.78 | 0.99 | 97.39 | 0.16 | 42.99 | 0.63 | 74.84 |
| Baseline-I | 0.18 | 55.78 | 0.74 | 87.74 | 0.73 | 78.90 | 0.5 | 74.91 |
| XMem++ | 0.48 | 71.03 | 0.87 | 95.48 | 0.16 | 31.11 | **0.69** | 80.65 |
| Ours | 0.37 | 69.91 | 0.98 | 96.27 | 0.75 | 78.91 | **0.69** | **82.14** |

Table 7: Quantitative results on a subset of PUMaVOS dataset (Bekuzarov et al. (2023)). At $k = 1$, we achieve higher quality in terms of $\mathbf{F}_{measure}$ and mIoU, outperforming XMem++. For other settings, we experience a small margin of loss, which is acceptable given that XMem++ is a fully supervised method, whereas our approach is a few-shot method.

| Methods | 1 frame | | 5 frames | | 10 frames | |
|---|---|---|---|---|---|---|
| | $\mathbf{F}_{meas.}$ | mIoU | $\mathbf{F}_{meas.}$ | mIoU | $\mathbf{F}_{meas.}$ | mIoU |
| Full Face 1 (XMem++) | 0.71 | 90.75 | 1.0 | 98.78 | 1.0 | 99.01 |
| Full Face 1 (Ours) | 0.98 | 96.46 | 0.99 | 96.76 | 1.0 | 96.73 |
| Full Face 2 (XMem++) | 0.80 | 89.92 | 0.96 | 96.64 | 0.97 | 97.35 |
| Full Face 2 (Ours) | 0.85 | 90.38 | 0.91 | 93.10 | 0.93 | 93.78 |
| Chair (XMem++) | 0.99 | 95.72 | 1.0 | 96.57 | 1.0 | 96.65 |
| Chair (Ours) | 0.32 | 63.32 | 0.98 | 90.62 | 0.99 | 89.82 |
| Half Face 1 (XMem++) | 0.82 | 90.52 | 0.94 | 94.54 | 0.96 | 95.49 |
| Half Face 1 (Ours) | 0.55 | 79.75 | 0.92 | 90.69 | 0.93 | 91.37 |
| Half Face 2 (XMem++) | 0.48 | 71.03 | 0.77 | 87.87 | 0.85 | 91.41 |
| Half Face 2 (Ours) | 0.37 | 69.91 | 0.66 | 81.06 | 0.83 | 87.17 |
| Long Scene Scale (XMem++) | 0.87 | 95.48 | 0.99 | 98.36 | 1.0 | 98.91 |
| Long Scene Scale (Ours) | 0.98 | 96.27 | 1.0 | 96.87 | 1.0 | 96.79 |
| Vlog (XMem++) | 0.16 | 31.11 | 0.55 | 62.84 | 0.82 | 82.52 |
| Vlog (Ours) | 0.75 | 78.91 | 0.86 | 84.01 | 0.90 | 85.29 |
| Mean (XMem++) | **0.69** | 80.65 | 0.89 | **90.80** | **0.94** | **94.48** |
| Mean (Ours) | **0.69** | **82.14** | **0.90** | 90.44 | **0.94** | 91.56 |

**Joint training vs two-phases training.** We employ two-phase training instead of joint training due to its increased stability and general improved performance. We have ablated our training recipe on SMITE-50 dataset in this experiment. Based on the results in Tab. 8, it is evident that two phase training gives us better performance quantitatively. In addition to this, we performed Levene's test on the training strategies to asses the training stability of both. We observed a significant difference (p = 0.0036) between the variance of losses in joint training and two-phase training. This ensures that two-phase training is more stable which can also be reflected visually in Fig 9 which confirms our findings. In both the cases, the number of epochs and the training time is same.

Table 8: Ablation of joint vs two-phase training on SMITE-50.

| Training Strategy | mIOU | Training Time |
|---|---|---|
| Joint | 62.17 | 20 mins |
| **Two-Phase** | **75.14** | **20 mins** |

**Cross-category generalization.** SMITE is capable of of segmenting objects that differ from the training samples. For example, when trained on horses, it is able to perform an acceptable segmentation on camels and giraffes as shown in Fig 10(a). For intra-class categories like in car category, even if we train SMITE with reference images of Sedan car, our model can still segment SUV car which is structurally varying.

**Multi-Object Flexible Granularity.** SMITE supports multi-object flexible granularity. For instance, when presented with a training image containing both a horse and a car, the model learns distinct representations for each object simultaneously as shown in Fig 11. This enables the model to generalize effectively, handling multiple objects with varying levels of detail in a cohesive manner.

## 7.3 EXTRA QUANTITATIVE RESULTS

**Performance on PUMaVOS and comparison with XMem++.** Here, we provide a comparison with XMem++ on their proposed dataset PUMaVOS to show that our method is effective on other datasets than our proposed SMITE-50. For this evaluation, we considered the following seven video splits namely :*Chair*, *Full face 1*, *Full face 2*, *Half face 1*, *Half face 2*, *Long Scene scale* and *Vlog* respectively. We chose the following categories as the object parts in these videos are not too small and also have flexible granularity. Note that XMem++ is a semi-supervised video segmentation technique meaning that it has been trained on large scale video segmentation datasets such as DAVIS or YoutubeVOS (Xu et al., 2018b). Nevertheless, SMITE performs comparably and even outperforms

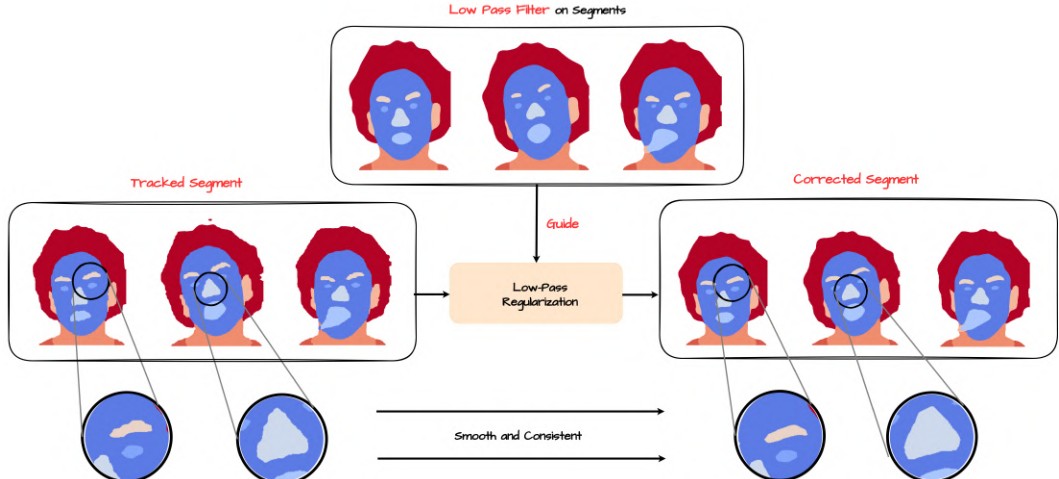

Figure 8: **Importance of using LPF** While applying tracking it leads to segments which has jagged edges and pixel drifts. Using LPF on the segments, treats it as a guide which corrects the inconsistent segments after tracking into more spatially consistent segments which preserves the original structure.

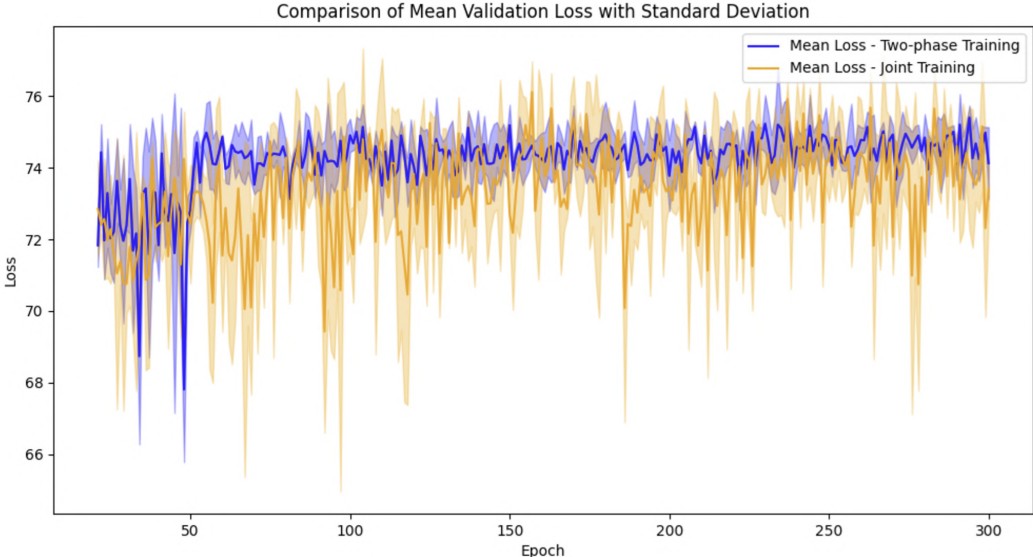

Figure 9: Two phase training highlighted as blue shows significant less oscillation in training and provides more stability.

it in case of working with fewer frames (e.g., a single shot; see Tab. 6). Predictably, when XMem++ is provided with more frames per video (e.g., 10), its performance slightly exceeds ours since SMITE is not trained on a large segmentation dataset. However, with a smaller number of frames (e.g., one or five), our method either outperforms or matches the performance of XMem++. See Tables 6 and 7, for the quantitative comparisons.

**Performance on DAVIS Dataset.** Note that the strength of our method is to perform arbitrary granularity and multi-granular segmentation and DAVIS only contains single objects rather than different granularity levels. To further contrast our method to SOTA for single object segmentation, we evaluate our dataset on DAVIS-2016 for the video object segmentation (VOS) task using two different settings : a) Zero-Shot VOS in Table 9 and b) Semi-Supervised VOS in Table 10 respectively. The first setting follows a zero-shot approach, where we train our model on one randomly selected image per video from the training set and report the results. However, our method has a limitation—it struggles with small objects. This is a common limitation for most existing diffusion based segmentation

Table 9: **Quantitative evaluation on DAVIS-2016 in a zero-shot setting.** Our method excels in multi-granular segmentation, but DAVIS-2016 only contains single objects. To compare with SOTA methods, we evaluate DAVIS-2016 in a zero-shot setting, training on one image per video from the training set. The results are reported as SMITE on the entire dataset and as SMITE$^+$ on the dataset excluding two samples (2/20) that are small and unsuitable for our method.

| Methods | J↑ | F↑ |
|---|---|---|
| FSNet (Ji et al., 2021) | 83.4 | 83.1 |
| F2Net (Liu et al., 2021) | 83.1 | 84.4 |
| TransportNet (Zhang et al., 2021) | 84.5 | 85.0 |
| AMCNet (Yang et al., 2021a) | 84.5 | 84.6 |
| RTNet (Ren et al., 2021) | 85.6 | 84.7 |
| CFANet (Chen et al., 2022) | 83.5 | 82.0 |
| D2Conv3D (Schmidt et al., 2022) | 85.5 | 86.5 |
| IMPNet (Lee et al., 2022) | 84.5 | 86.7 |
| DBSNet (Fan et al., 2022) | 85.9 | 84.7 |
| HFAN (Pei et al., 2022) | 86.2 | 87.1 |
| TMO (Cho et al., 2023) | 85.6 | 86.6 |
| OAST (Su et al., 2023) | 86.6 | 87.4 |
| **SMITE** | 80.8 | 82.9 |
| **SMITE$^+$** | 82.39 | **87.6** |

Table 10: **Quantitative evaluation on DAVIS-2016 in a semi-supervised setting.** Our method is designed for multi-granular segmentation, but DAVIS-2016 only contains single objects. To compare with SOTA methods, we evaluate DAVIS-2016 in a semi-supervised setting, training on one image per video from the validation set. The results are reported as SMITE on the entire dataset and as SMITE$^+$ on the dataset excluding two samples (2/20) that are small and unsuitable for our method.

| Method | J↑ | F↑ |
|---|---|---|
| STM (Oh et al., 2019) | 88.7 | 89.9 |
| AFB-URR (Liang et al., 2020) | 88.3 | 90.5 |
| CFBI (Yang et al., 2020) | 88.9 | 88.7 |
| RMNet (Xie et al., 2021b) | 89.6 | 92.0 |
| HMMN (Seong et al., 2021) | 89.6 | 92.4 |
| MiVOS (Cheng et al., 2021a) | 90.8 | 92.5 |
| STCN (Cheng et al., 2021b) | 90.1 | 92.1 |
| JOINT (Mao et al., 2021b) | 90.4 | 92.7 |
| AOT (Yang et al., 2021b) | 91.1 | 92.1 |
| XMem (Cheng & Schwing, 2022) | 91.5 | **92.7** |
| **SMITE** | 83.4 | 80.7 |
| **SMITE$^+$** | **92.0** | 88.7 |

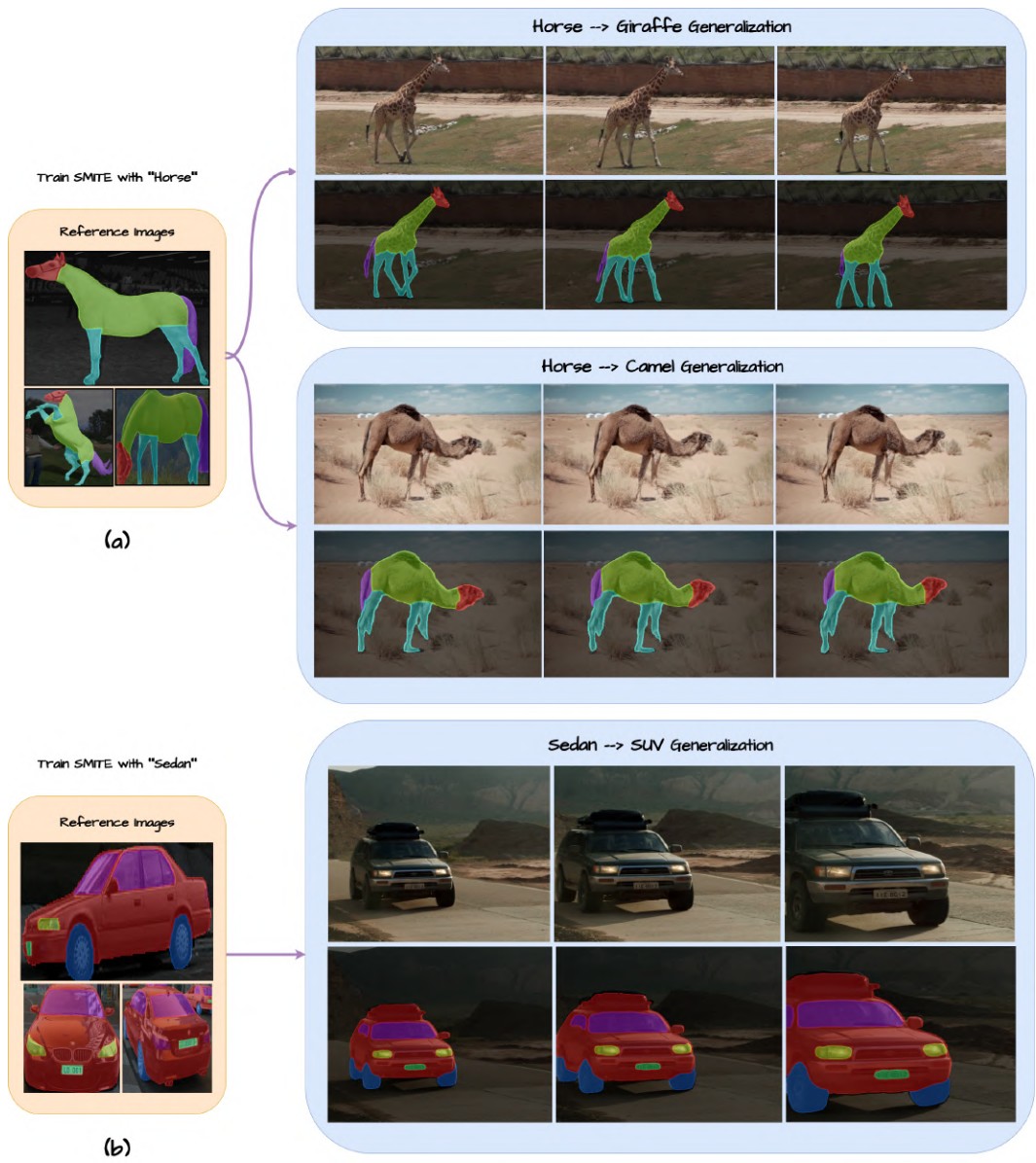

Figure 10: **Cross-Category Generalization** SMITE supports cross-category generalization where dispite of training with reference images of (a) *horse* category, our model can successfully generalize to *giraffe,camel* which has large topological changes and training with reference images of (b) *sedan* car, our model can successfully generalize to *SUV* car which has intra-category variations.

approaches Namekata et al. (2024); Khani et al. (2024). Additionally, two out of the 20 videos contain thin lines to be segmented (Fig 13), making them unsuitable for segmentation using our method. Therefore, in addition to reporting numbers on the entire test-set, we also exclude these two videos and report the average performance (SMITE+) on the remaining DAVIS-2016 videos.

Secondly, we evaluate our method in a semi-supervised segmentation setting, where only a single frame is used for each test video.While many models, both in zero-shot and semi-supervised settings, are specifically trained for video segmentation using videos from datasets like YouTubeVOS or DAVIS, our approach requires only a few frames for training. Despite this minimal requirement, we achieve competitive performance on most examples, except in cases involving small objects, where our method faces limitations.

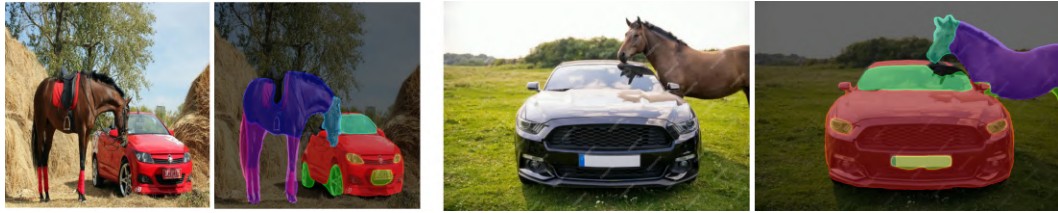

Figure 11: **Multi-Object Segmentation.** Our method is capable of multi-object segmentation.

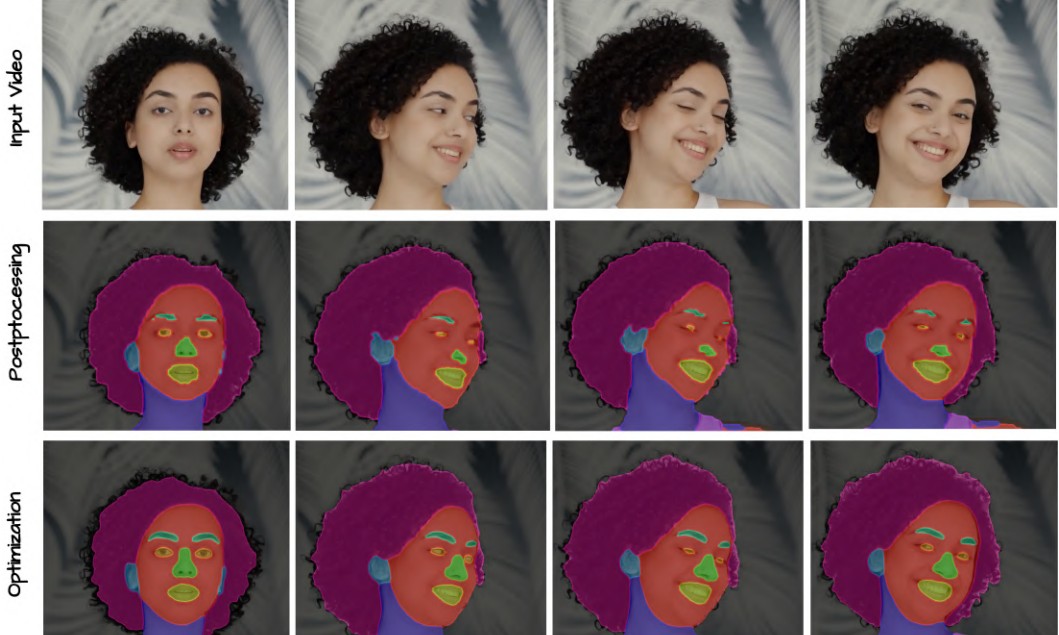

Figure 12: **Importance of applying Tracker.** We ablated the usage of tracking algorithm in two scenarios : (a) During postprocessing (second row) and (b) During inference optimization (last row). From the figure, it can be observed that applying tracker on the segments $S_{was}$ during postprocessing (Baseline-II) leads to segment drifts thus resulting in jagged edges. Such problems can be mitigated if we use tracking during inference optimization (Eq. 5) stage since the optimization process slowly propagates the corrections across the entire video during the denoising stage, thus resulting in lesser segment drift and temporally consistent predictions.

### 7.4 EXTRA QUALITATIVE RESULTS

For more challenging cases, we include more results in Fig 14. In one instance, the ice cream cone is occluded by a paper napkin, and the ice cream itself is obscured and blended by a face, yet SMITE is still able to generate correct results. Furthermore, the turtle nearly blends with the background in terms of color and visual patterns, but our method successfully tracks the segments.

We have included qualitative comparison videos in the supplementary file to provide more examples in video format. The actual videos are in higher resolution, we have compressed the video to upload it on the website easily.

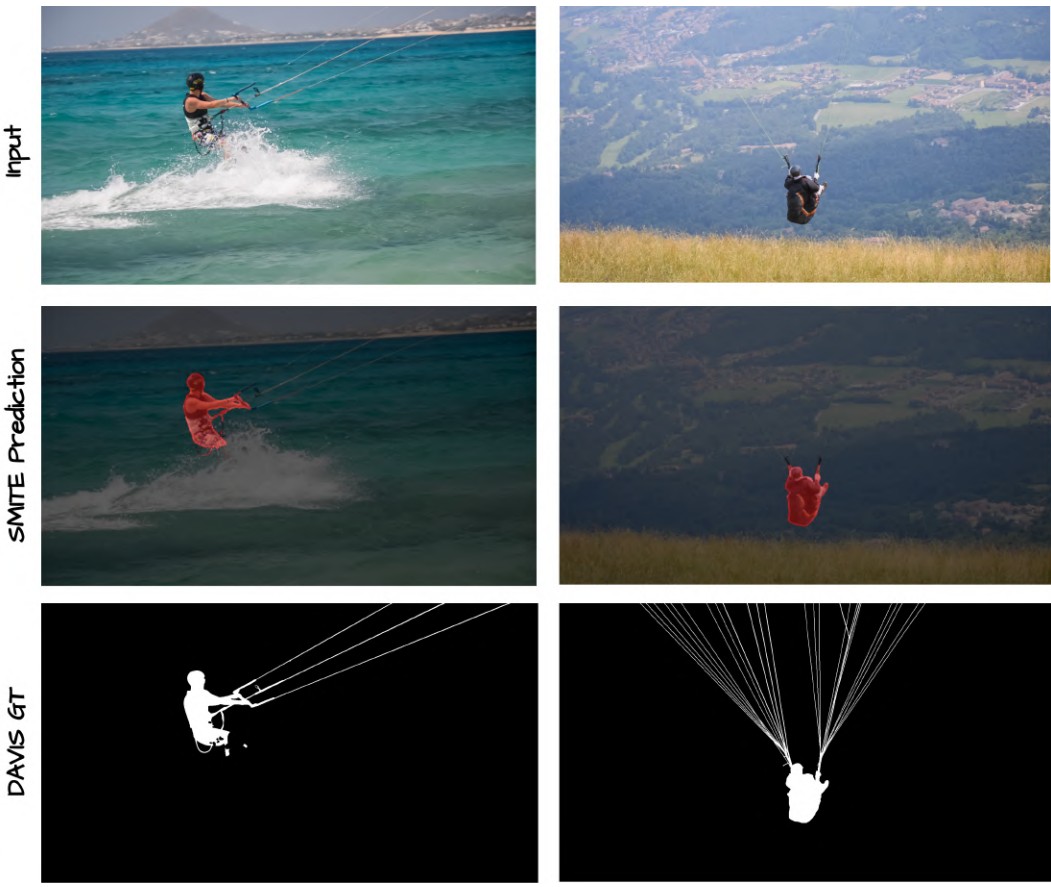

Figure 13: **DAVIS Test Set Visualization** SMITE segments poorly in scenes which has thin structures like *strings* in both the scenes.

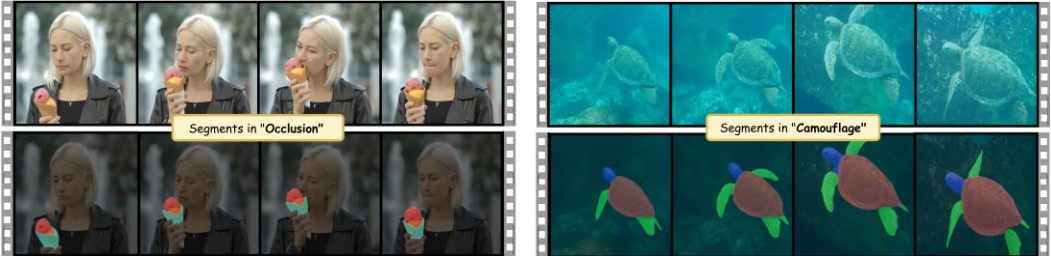

Figure 14: **Segmentation results in challenging scenarios .** SMITE accurately segments out the objects under occlusion ("ice-cream") or camouflage ("turtle") highlighting the robustness of our segmentation technique.

## 7.5 IMPLEMENTATION DETAILS AND DESIGN CHOICES EXPLANATION

All experiments were conducted on a single NVIDIA RTX 3090 GPU. During the learning phase, we initially optimized only the text embeddings for the first 100 iterations. For the subsequent iterations, we optimized the cross-attention to_k and to_v parameters.

For the categories of horses, cars, faces and non-text, we used 10 reference images from our SMITE-50 benchmark for both our method and the baseline comparisons.

Regarding the window size in our tracking module, we found that fast-moving objects benefit from a smaller window size to mitigate potential bias. Consequently, we set the window size to 7 for horses

and 15 for other categories. To better work on smaller objects, we employ a two-step approach. First, our model generates an initial segmentation estimate. We then crop the image around this initial estimate and reapply our methods to obtain a more fine-grained segmentation.

During inference, we added noise corresponding to 100 timesteps and performed a single denoising pass when segment tracking and voting were not employed. When using segment tracking and voting, we applied spatio-temporal guidance at each denoising step and conducted backpropagation 15 times per denoising timestep. For the regularization parameters, we set $\lambda_{\text{Reg}}$ across all experiments. The tracking parameter $\lambda_{\text{Tracking}}$ was set to 1 for horses, 0.5 for faces, and either 0.2 or 1 for cars. Additionally, we applied a Discrete Cosine Transform (DCT) low-pass filter with a threshold of 0.4.

**Pseudo Code of Temporal Voting**

---

**Algorithm 1** Temporal Voting

---
1: Input: $X$: a pixel at frame $t$, $W$: window size
2: $X_s \leftarrow$ Correspondence of $X$ at frame $s$ (obtained by $\text{CoTracker}(X, s)$)
3: $\text{Vis}(X_s, s)$: visibility of $X_s$ (obtained by CoTracker)
4: $\text{Visible\_Set} \leftarrow \{i \in \text{range}\left(-\frac{W}{2}, \frac{W}{2}\right)$ **if** $\text{Vis}(X_{s_i}) == 1\}$
5: $\mathbf{P} \leftarrow \text{Most\_Occurrence}\left(\text{S}(X_i).\text{argmax}(\text{dim} = 0)\right)$ where $i \in \text{Visible\_Set}$
6: $\text{total} \leftarrow 0$, $\text{count} \leftarrow 0$
7: **for all** $p \in \text{Visible\_Set}$ **do**
8:     **if** $\text{S}(X_i).\text{argmax}(\text{dim=0}) == P$ **then**
9:        $\text{total} \leftarrow \text{total} + \text{S}(X_i)$
10:        $\text{count} \leftarrow \text{count} + 1$
11:     **end if**
12: **end for**
13: $\text{S}_{\text{tracked}}(X) \leftarrow \dfrac{\text{total}}{\text{count}}$

---

**Bidirectional Tracking and Resolution Reduction in CoTracker** Note that points queried at different frames are tracked incorrectly before the query frame. This is because CoTracker is an online algorithm and only tracks points in one direction. However, we can also run it backward from the queried point to track in both directions. So by setting $backward\_tracking$ to True we are able to track points in both directions which is crucial for our voting mechanism.

**Time and Memory Analysis.** Using 10 training images with a batch size of 1 on 15 GB of GPU VRAM, training takes 20 minutes. By increasing the batch size and utilizing more GPU VRAM, training time can be reduced to 7 minutes. Although we use 300 epochs, the best accuracy is often reached before epoch 200, so training time can be further decreased. Training time is unaffected by whether we learn the text embedding or U-Net weights. For inference, each frame takes 26 seconds and requires 60 GB of GPU VRAM to process the entire video. However, it is possible to adjust settings to use only 15 GB, though this increases the inference time.

**Long video processing.** A key goal for us is to adapt our method to work efficiently on smaller GPUs. Unlike most video editing techniques that require high-end hardware like an A100 GPU and typically handle up to 24 frames, our approach aims for broader applicability. We identified that gradient computation during inference optimization is particularly demanding on resources. To address this, we segment the latent space into smaller windows—for example, (1 to k), (k+1 to 2k), (2k+1 to 3k), and so on—across different timesteps and optimize each window independently. This segmentation has proven not to compromise the final results. Additionally, for tasks such as segmenting 1,000 frames (PUMAVOS samples) consistently, our method processes batches of 2×T frames at a time (you can set T based on your GPU's capacity). We then save the states of the last (T+1) frames and replace them with the reference state in the next iteration. These strategies enable our model to process longer videos effectively on GPUs with as little as 24 GB of VRAM.

**Enhanced Convergence.** The strategy to accelerate convergence and simplify parameter tuning in the code involves the use of an Adam-like optimization approach that dynamically adapts the learning rate and gradient updates for the latent variables. Specifically, the code implements the first and second moment estimates, denoted as $M1$ and $M2$, which accumulate the gradients and squared gradients, respectively.

In each iteration, the first moment estimate $M1$ captures the exponentially weighted average of the gradients, while the second moment estimate $M2$ tracks the squared gradients. These moment estimates are then bias-corrected by dividing them by $1 - \beta_1^{t+1}$ and $1 - \beta_2^{t+1}$, where $\beta_1$ and $\beta_2$ are the momentum parameters typically set to 0.9 and 0.999, respectively. This bias correction ensures that the estimates are unbiased, particularly in the initial steps of optimization.

The learning rate $\alpha_t$ is dynamically scaled based on these corrected moment estimates, where the update step for the latent variables is computed as:

$$z_t' \leftarrow z_t - \alpha_t \cdot \frac{M1_{\text{corrected}}}{\sqrt{M2_{\text{corrected}}} + \epsilon}$$

This adaptive approach allows the optimizer to adjust the learning rate on a per-parameter basis, depending on the variance of the gradients, leading to faster convergence. By using this method, the optimizer can take larger steps when the gradients are consistent and smaller steps when they are noisy, which helps in avoiding overshooting or getting stuck in local minima. The combination of momentum-based updates and dynamic learning rate scaling makes the optimization process more robust, reducing the need for extensive manual tuning of hyperparameters such as the learning rate, and enabling more efficient convergence.

**Bidirectional tracking.** We use bidirectional tracking instead of unidirectional tracking for two main reasons. First, to manage longer videos, we implement a slicing approach where the last frames of the first slice are retained in the second slice to ensure continuity. Bidirectional tracking speeds up consistency between slices by allowing new frames in the second slice to directly reference frames from the first slice, while unidirectional tracking delays this process due to the need for updates to propagate. Second, tracking methods often struggle with fast-moving objects, as accuracy decreases with distance from the query pixel, risking loss of track. Bidirectional tracking enhances robustness in these situations. Additionally, in cases of occlusion, unidirectional tracking may fail without visible pixels to propagate information. Bidirectional tracking mitigates this by leveraging data from both past and future frames, maintaining accuracy even during occlusions.

### 7.6 HOW DOES WAS ATTENTION WORK?

SLiMe refines latent diffusion models by learning special text embeddings that guide segmentation using the *Weighted Accumulated Self-Attention* (WAS) map. In each denoising step, a latent diffusion U-Net uses cross-attention to link latent features with text tokens, and self-attention to capture global spatial information. SLiMe combines these two forms of attention through WAS, enabling it to identify clear boundaries and consistent object regions, even in unseen images.

**Weighted Accumulated Self-Attention.** Let $\boldsymbol{S}_{ca}^{\ell}$ be the cross-attention map and $\boldsymbol{S}_{sa}^{\ell}$ be the self-attention map at layer $\ell$. We first average resized cross-attention maps:

$$\boldsymbol{A}_{ca} = \text{Average}_{\ell}\Big(\text{Resize}\big(\{\boldsymbol{S}_{ca}^{\ell}\}\big)\Big),$$

and do the same for self-attention maps:

$$\boldsymbol{A}_{sa} = \text{Average}_{\ell}\big(\{\boldsymbol{S}_{sa}^{\ell}\}\big).$$

Next, we resize $\boldsymbol{A}_{ca}^{k}$ (the map for the $k$-th text embedding) to match the shape of $\boldsymbol{A}_{sa}$:

$$\boldsymbol{R}_{ca}^{k} = \text{Resize}\big(\boldsymbol{A}_{ca}^{k}\big),$$

then flatten $\boldsymbol{R}_{ca}^{k}$ and multiply it element-wise by $\boldsymbol{A}_{sa}$. Summing the result across channels gives the WAS map:

$$\boldsymbol{S}_{\text{WAS}}^{k} = \sum\Big(\text{flatten}\big(\boldsymbol{R}_{ca}^{k}\big) \odot \boldsymbol{A}_{sa}\Big).$$

This step combines the text-based localization (from cross-attention) with the detailed spatial information (from self-attention), leading to sharper and more meaningful segmentations.

**Loss Functions.** The text embeddings are optimized using three loss functions. The cross-entropy loss encourages the cross-attention maps to align with the ground truth mask $M$:

$$L_{\text{CE}} = \text{CE}(\boldsymbol{S}_{ca}, M).$$

The MSE loss on the WAS map aligns self-attention with the ground truth:

$$L_{\text{MSE}} = \sum_k \left\|\text{Resize}(\boldsymbol{S}_{\text{WAS}}^k) - M_k\right\|_2^2.$$

Finally, the diffusion-based regularization term keeps the learned embeddings within the pretrained latent diffusion distribution:

$$L_{\text{LDM}} = \mathbb{E}_{\boldsymbol{x}, \epsilon, t}\left[\left\|\epsilon - \epsilon_\theta(\boldsymbol{z}_t, t, y)\right\|_2^2\right].$$

The total loss is:

$$L = L_{\text{CE}} + \alpha L_{\text{MSE}} + \beta L_{\text{LDM}},$$

where $\alpha$ and $\beta$ control how strongly we emphasize spatial accuracy versus staying consistent with the original diffusion model. Once trained, SLiMe's embeddings can segment new images at similar granularity without extra fine-tuning.

## 7.7 IMAGE SEGMENTATION RESULTS

We tested our method on an image dataset (e.g PASCAL-Part(Chen et al., 2014)) to demonstrate the enhancements achieved through modifications to our architecture and optimization. As shown in Tables 11 and 12, our approach shows significant improvement over SLiMe even for image segmentation for car and horse split of PASCAL-Part dataset. This highlights the effectiveness of our design choices in SMITE, particularly the cross-attention tuning.

Table 11: **Image segmentation results for class car** SMITE consistently outperforms SLiMe. The first two rows show the supervised methods, for which we use the reported numbers in ReGAN. The second two rows show the methods with 1-sample setting and the last three rows refer to the 10-sample setting methods. $\star$ indicates the supervised methods.

|  | Body | Light | Plate | Wheel | Window | Background | Average |
|---|---|---|---|---|---|---|---|
| CNN$^\star$ | 73.4 | 42.2 | 41.7 | 66.3 | 61.0 | 67.4 | 58.7 |
| CNN+CRF$^\star$ | 75.4 | 36.1 | 35.8 | 64.3 | 61.8 | 68.7 | 57.0 |
| SegGPT (Wang et al., 2023)$^\star$ | 62.7 | 18.5 | 25.8 | 65.8 | 69.5 | 77.7 | 53.3 |
| OIParts (Dai et al., 2024) | 77.7 | **59.1** | **57.2** | 66.9 | 59.2 | 71.1 | 65.2 |
| ReGAN (Tritrong et al., 2021) | 75.5 | 29.3 | 17.8 | 57.2 | 62.4 | 70.7 | 52.15 |
| SLiMe (Khani et al., 2024) | 81.5 | 56.8 | 54.8 | 68.3 | 70.3 | 78.4 | 68.3 |
| Ours | **82.3** | 57.5 | 55.9 | **70.1** | **72.6** | **80.1** | **69.8** |

Table 12: **Image segmentation results for class horse.** SMITE outperforms ReGAN, OPParts, SegDDPM, SegGPT and SLiMe on average and most of the parts. The first two rows show the supervised methods, for which we use the reported numbers in ReGAN. The middle two rows show the 1-sample setting, and the last four rows are the results of the 10-sample settings. $\star$ indicates the supervised methods.

|  | Head | Leg | Neck+Torso | Tail | Background | Average |
|---|---|---|---|---|---|---|
| Shape+Appereance$^\star$ | 47.2 | 38.2 | 66.7 | - | - | - |
| CNN+CRF$^\star$ | 55.0 | 46.8 | - | 37.2 | 76 | - |
| SegGPT (Wang et al., 2023)$^\star$ | 41.1 | 49.8 | 58.6 | 15.5 | 36.4 | 40.3 |
| OIParts (Dai et al., 2024) | **73.0** | 50.7 | 72.6 | **60.3** | 77.7 | 66.9 |
| ReGAN (Tritrong et al., 2021) | 50.1 | 49.6 | 70.5 | 19.9 | 81.6 | 54.3 |
| SegDDPM (Baranchuk et al., 2021) | 41.0 | 59.1 | 69.9 | 39.3 | **84.3** | 58.7 |
| SLiMe (Khani et al., 2024) | 63.8 | 59.5 | 68.1 | 45.4 | 79.6 | 63.3 |
| Ours | 64.5 | **61.9** | **73.2** | 48.1 | 83.5 | **66.2** |

