# OpenReview forum: "SMITE: Segment Me In TimE"
_ICLR.cc/2025/Conference — ICLR 2025 Poster_

### Official Review · Reviewer_askr · 2024-10-27

**Soundness:** 3
**Presentation:** 3
**Contribution:** 2
**Rating:** 6
**Confidence:** 4

**Summary:**

This paper presents SMITE, a few-shot video segmentation method with fine granularity tracking. SMITE is based on SLiMe, a few-shot image segmentation method that finetunes text tokens for a diffusion network and probes attention masks to output a mask.  SMITE additionally introduces cross-attention finetuning of the diffusion network, a segment tracking and voting module based on point tracking, and a low-pass regularization term. SMITE achieves better segmentation performance than applying baseline SLiME per frame in a newly proposed dataset SMITE-50.

**Strengths:**

This paper tackles the hard yet useful task of few-shot, fine-granularity video segmentation. This task has potentially wide applications. I think the main contribution of this work is taking the initial crack at this problem with reasonable performance in segmentation quality and temporal stability when compared to applying the SLiME per frame.

Technique-wise, segmentation tracking/voting and low-pass regularization are reasonable in handling temporal flickering unique to videos, and the results also supported this.

The proposed dataset SMITE-50 is also potentially useful for future work, as it contains densely labeled segmentation which is a rare type of data.

**Weaknesses:**

I don’t see it mentioned in the paper, but I suspect that SMITE might be too slow for most practical applications. In addition to finetuning the text tokens and the cross-attention layers during a training stage, SMITE additionally performs dense, windowed point tracking – there seem to be a lot of windows (linear to video length) and a lot of repeated computation in the point tracker (linear to windows size). SMITE also minimizes an energy function which is also costly. Besides, given that SMITE expands the spatial self-attention to the spatiotemporal volume, it would be quite memory intensive for longer videos.

I think the authors can also compare SMITE to existing few-shot video object segmentation methods like [Multi-grained Temporal Prototype Learning for Few-shot Video Object Segmentation, ICCV 2023]. This can happen in two dimensions: 1) applying few-shot VOS to the fine-grained segmentation setting, and 2) applying SMITE to the few-shot VOS setting. This can highlight whether the fine-grained segmentation setting significantly differs from the object setting and supplements the existing sparse baselines.

In Table 2, should there be a row with neither of the loss and another row with only the low-pass loss? Also, can the ablations be run on the entire SMITE-50 dataset rather than just on the CARS subset? This would make the results more significant.

**Questions:**

What is the speed and memory requirement of the proposed method? How many frames can it process?

What if an inflated U-Net is not used during text embedding optimization (e.g., the segment prompts are just images, not videos) but only used during inference?

---

> ### Author Response · Authors · 2024-11-26
> **Official Comment by Authors**
>
> Thank you for appreciating the importance of our work and providing insightful feedback.
>
> **W1: "time and memory analysis" :** We have included discussions on *“Time and Memory Analysis”* in Section 7.3. For longer video processing, please refer to our discussion in Section 7.5 of Appendix.
>
> **W2: "comparison with few-shot VOS" :** Thanks for the suggestion ! Kindly refer to Section 5 and Tables 5 and 6 where we have already included comparisons of our approach in few-shot setting with existing baselines. In addition to this, we have now added *“Comparison on DAVIS”* in Section 7.3 of Appendix to include additional experiments on DAVIS dataset.
>
> **W3: "limited ablation study" :** Thanks! We have now updated Table 2 with detailed network ablation on entire SMITE-50 dataset and provided related discussion in L455-463. It is to be noted, that the low-frequency regularization (LP Reg) helps maintain spatial structure and prevents segment drift when tracking is applied to $S_{WAS}$. Without tracking, LP Reg has minimal effect since no segment drift occurs in WAS maps (thus change of energy in $E_{reg}$ is almost zero), so we do not ablate LP Reg separately without tracking. Additionally, we have also included ablations in appendix Section 7.2.
>
> **Q1: "time and memory analysis and any frame limitation?" :** Please refer to the general response above and the *“Time and Memory Analysis”* section. We do not have any frame processing limit as we split the video into chunks for longer videos. Please refer to Section 7.5 of Appendix for further details.
>
> **Q2: "no inflated U-Net during training" :** Very good question! Using standard Unet during training and Inflated-Unet during testing will not impact our results. This is because our training is composed of single images and we process them as videos of length one. Thus, it never utilizes the temporal attention in the inflated UNet. Therefore, using inflated Unet or standard Unet during training makes no difference to our final results.

---

> ### Comment · Reviewer_askr · 2024-11-27
>
> I thank the authors for the detailed reply and updates to the paper.
>
> W1: "time and memory analysis" -- "We have included discussions on “Time and Memory Analysis” in Section 7.3" -- do you mean Section 7.5? This confirms my suspicion that the algorithm is slow but this is fine. How does this running time and memory usage compare to existing methods?
>
> "However, it is possible to adjust settings to use only 15 GB, though this increases the inference time." -- by how much?
>
> For long video processing (next paragraph) -- "This segmentation has proven not to compromise the final results" -- proven by whom?
>
> I have no further questions on other aspects.

---

> > ### Author Response · Authors · 2024-11-27
> >
> > **Q1: "Time comparison with other methods":** Thank you for pointing out the error; we meant Section 7.5, not Section 7.3.
> > Regarding the runtime comparison, the inference times and mIOU scores on the SMITE-50 dataset for various methods are as follows:
> >
> > * Baseline-I: Approximately 1–2 seconds per frame; mIOU of 66.62.
> >
> > * Baseline-II: Over 100 seconds per frame (due to tracking on the pixel space); mIOU of 65.50.
> >
> > * Baseline-III: Approximately 1–2 seconds per frame; mIOU of 69.36.
> >
> > * GSAM2: Less than 2 seconds per frame; mIOU of 66.53.
> >
> > In comparison, our method achieves a significantly higher mIOU of 75.14, albeit with a longer inference time due to an optimization step during processing. It is important to note that inference time can vary depending on specific use cases, as some applications may prioritize higher accuracy over faster runtime.
> >
> > Additionally, FRESCO [1], a video editing method based on diffusion models, has an inference time of 9–10 seconds per frame. This highlights that longer inference times are often expected with diffusion-based approaches. Reducing the inference time of our method could be addressed in future work.
> >
> > **Q2: "Inference time’s increase with 15 GB settings":**  Adjusting the settings to reduce memory usage to 15 GB increases the inference time by approximately 50%.
> >
> > **Q3: "Clarify (proven)":** Thank you for pointing this out. We meant that after making this modification, we did not observe any significant visual differences in the results. We will revise the sentence to: "Based on our visual assessments, this segmentation has not compromised the final results."
> >
> > [1] S. Yang et al., "FRESCO: Spatial-Temporal Correspondence for Zero-Shot Video Translation," CoRR, 2024.

---

> > > ### Comment · Reviewer_askr · 2024-11-28
> > >
> > > Thank you for the response. I have no other questions.
> > > I will keep my original rating, as that rating was conditional on a satisfactory response.

---

> > > > ### Author Response · Authors · 2024-11-28
> > > >
> > > > Thank you for your time and positive feedback. We are glad that our response addressed your concerns.

---

### Official Review · Reviewer_K95K · 2024-10-29

**Soundness:** 3
**Presentation:** 3
**Contribution:** 3
**Rating:** 8
**Confidence:** 3

**Summary:**

This work presents a framework for segmenting videos into parts at adjustable levels of granularity, using only a single or few annotated images as references. The framework comprises several main components:
First, it optimizes text embeddings and fine-tunes a stable diffusion model so that the WAS attention maps (a combination of self and cross-attention maps) align with the annotated reference images.
Next, temporal consistency is enforced through specific loss terms, enabling the framework to generate segmentation maps from the input video using the optimized text embeddings and fine-tuned diffusion model.

**Strengths:**

Novel and interesting problem formulation.
The motivations behind the design choices are presented clearly.
Qualitative results seem decent.
A new video part-level dataset and SOTA performance.

**Weaknesses:**

1. Limited competitors - The baseline (frame-by-frame) might be a bit too simplistic.
Maybe a better baseline would be a simple video extension of SLIME (without fine-tuning the cross-attention layers). This video extension is pretty trivial and should have some consistency.
Another baseline could be using SLIME (single frame) with tracking.
2. Limited ablation study -
I would have liked to know what is the contribution of each step.
In addition, I think more ablation would make the paper stronger. For example, how similar should be the input object to the one in the video? Would horse segmentation work on zebras? and on dogs?

In addition: Table 2 only states what happens if only the regularization step is dropped. What about dropping the tracking? And what about dropping both?

**Questions:**

1. The differences in Figure 3 could be better presented (the difference between (c) and (d) is hardly visible).

2. It was slightly challenging to follow all the technical details.
Few small modifications might reduce the reading time:
For example:
a. In section 4.2 3 loss terms were mentioned and in section 4.4 2 other loss terms were mentioned; It will be clearer if the authors stated clearly how and in what phase each loss term is used (all together? two-phase optimization?).
b. lines 267-268, sliding window -> temporal sliding window

3. Small technical note:
line 148: should the epsilon_theta after UNet be deleted?

4. Lines 303-305:
It is not clear how this scenario is different from using only cross attention maps.
If I understand what the authors suggested correctly: average voting can be computed directly on the masks as a "tracking post-process step" without the need for an optimization for tracking.
If so, why can't you simply calculate the average vote on the S_WAS as well?
If not, what was your intention?

5. Line 105 - you mention that some parts cannot be described by text.
Yet, you do find text embeddings that describe them, although it is reasonable to claim that these embeddings do not represent natural text descriptions.
However, it will be interesting to acknowledge/discuss this distinction.
Why does it make sense to find text embeddings for parts that cannot be described by texts?

---

> ### Author Response · Authors · 2024-11-26
> **Official Comment by Authors**
>
> Thank you for appreciating the importance of our work and providing insightful feedback.
>
> **W1: "video extension of SLiMe (no fine-tuning) - SLiMe (single frame) + tracking" :** Please see the general response above. As per your suggestion, we have now included two new baselines: a) Baseline-II is to use SLiMe and CoTracker simultaneously, and b) Baseline-III, which is the modification of SLiMe by only changing its UNet to the Inflated UNet. Our method outperforms both of these baselines in Table 1.
>
> **W2: "limited ablation study; input-object similarity analysis (e.g., horse → zebra, dog), low pass reg. step is dropped" :** We have now included detailed network ablation in Table 2. In addition, we have included ablation studies in Appendix Section 7.2.
> Additionally, we have added a new discussion in Section 7.2 on *“Cross-category generalization”* and added a visual result in Figure 10 where our model trained on horses can successfully generalize to camels and giraffe which has large topological changes compared to horses. The low-frequency regularization (LP Reg) helps maintain spatial structure and prevents segment drift when tracking is applied to $S_{WAS}$. Without tracking, LP Reg has minimal effect since no segment drift occurs in WAS maps (thus change of energy in $E_{reg}$ is almost zero), so we do not ablate LP Reg separately without tracking. We have updated it in L458-463.
>
> **Q1: "figure 3 differences (c vs. d) unclear" :** Kindly note that Figure 3 is a video when opened using Adobe Acrobat. In addition to this, we have provided a new video *“Ablation Video”* in the supplemental material to better show this.
>
> **Q2: "improve readability" :** Thank you for pointing this out. We have now corrected based on your suggestions and also corrected  lines L254-55 and L339. The loss terms used in Section 4.2 is used during training and in Section 4.4 is used during inference optimization. We have symbolically distinguished the training losses as $L$ and testing losses as $\varepsilon$.
>
> **Q3: "small technical note, $\epsilon_{\theta}$" :** Thank you for your suggestions, we have now addressed it in the revised paper.
>
> **Q4: "why not average voting on masks as post-process? - why not apply average voting on $S_{WAS}$?" :** Great question!
> Applying tracking and voting directly in the pixel space as post-processing, can lead to segmentation drift. This explains why Baseline II (SLiMe+CoTracker) which is added to the updated paper performs slightly worse than Baseline I. It is also visible in Figure 12 qualitatively. To address this, we have introduced a temporal voting mechanism on the attention maps enhanced with low-pass regularization within an optimization framework. Since this approach is quite effective, we did not test other possible voting scenarios that might be also promising. Hence we can treat it as one of our future work !
>
> **Q5: "justify using text embeddings for indescribable parts" :** That is an excellent question! Yes, for segments which are hard to explain with text, we learn embeddings corresponding to the visual segments, since such asymmetric segment descriptions are outside the text vocabulary. In fact, a word may not exist for a part but a text vector embedding through an optimization may be found which is not a regular readable text. This is similar to Textual Inversion [1] or other personalization techniques, where a known word for a particular image may not exist but an optimized vector embedding can represent a particular image.
>
> [1] Rinon Gal, et al., An image is worth one word: Personalizing text-toimage generation using textual inversion. arXiv 2022.

---

> > ### Comment · Reviewer_K95K · 2024-11-27
> >
> > I thank the authors for their detailed answer.
> > I will maintain my rating.

---

> ### Author Response · Authors · 2024-11-29
>
> Thank you very much! We appreciate your comments and positive feedback.

---

### Official Review · Reviewer_aCvu · 2024-10-31

**Soundness:** 3
**Presentation:** 3
**Contribution:** 2
**Rating:** 6
**Confidence:** 3

**Summary:**

This paper proposes a SMITE framework for segmenting object fragments across video frames. The method consists of denosing module, diffusion module and tracking module. Overall, it achieves satisfactory performance for video segmentation over various levels of granularity.

**Strengths:**

1)This proposed method combines information from text, tracker and spatial temporal features for video segmentation at arbitrary level.
2)SMITE achieves state of the art performance on the SMITE-50 dataset.

**Weaknesses:**

1)The proposed method is only compared with two methods in Table 1, while there are lots of few-shot and zero-shot segmentation methods.
Besides, it seems that the proposed method can be directly applied on multi-target video segmentation datasets like DAVIS2017. So, there should be more comprehensive comparison, for both methods and datasets.
2)Speed is important for online tracking; so there should be some runtime analysis.
3)Limitations of the method is not discussed in experiments.
4)How does the proposed method work on random videos and random annotation references?

**Questions:**

Same as weakness.

---

> ### Author Response · Authors · 2024-11-26
> **Official Comment by Authors**
>
> Thank you for appreciating the importance of our work and providing insightful feedback.
>
> **W1: "limited comparison (add few-shot, zero-shot methods)" :** Please refer to Section 5 and also Tables 5 and 6 where we had already included  comparisons on PumaVOS dataset in few-shot setting with Xmem++. In addition, we have now added Tables 8 and 9 on DAVIS-2016 dataset. Please see the general response above and refer to the *“Comparison on DAVIS”* in Section 7.3 of Appendix.
>
> **W2: "time analysis" :** We would like to clarify that we do not claim our method is capable of real-time segmentation in the paper. Please refer to the *"Time and Memory Analysis"* section in the Appendix of the paper and also the general response. Please also see our response to reviewer askr under Q1 for more details on time analysis.
>
> **W3: "limitations" :** We already had some limitations included in Section 6 in the original submission but we now have expanded the limitations to better clarify our method's shortcomings.
>
> **W4: "random videos, random references?" :** We do not use any particular method to hand pick or select videos or annotations. As long as references are of high quality with most of the necessary parts visible, our technique is able to perform reasonably well.

---

> > ### Author Response · Authors · 2024-12-01
> >
> > Dear reviewer, thank you once again for your valuable feedback. We made our best efforts to address your concerns, particularly adding few/zero shot comparisons, limitations, and time analysis to the paper. We believe these clarifications strengthened our paper and we hope they resolved the issues you raised. We would appreciate any further feedback you may provide in your evaluation of our submission.

---

### Official Review · Reviewer_rK61 · 2024-11-02

**Soundness:** 3
**Presentation:** 3
**Contribution:** 3
**Rating:** 8
**Confidence:** 3

**Summary:**

This paper presents SMITE (Segment Me In Time), a novel approach for one/few-shot video object part segmentation with flexible granularity. Building upon the image-based SLiMe method, SMITE leverages a pre-trained text-to-image diffusion model (Stable Diffusion) and introduces a tracking mechanism and low-pass filtering to maintain temporal consistency in video segmentations. Given one or a few annotated reference images, SMITE aims to segment unseen videos of objects from the same category, respecting the provided segmentation granularity. The authors introduce a new dataset, SMITE-50, to benchmark performance and compare against relevant baselines.

**Strengths:**

1. Originality : The proposed method innovatively combines pre-trained text-to-image diffusion models with a temporal tracking mechanism to achieve video segmentation with flexible granularity using only a few reference images.
1. Quality : The comprehensive experiments, including quantitative evaluations on the new SMITE-50 dataset, qualitative comparisons, ablation studies, and user studies, provide solid support for the method's effectiveness.
1. Clarity : The paper is well-structured, with clear explanations of the problem, methodology, and results. Figures and diagrams are used effectively to enhance understanding.
1. Significance: Addressing the challenge of few-shot video object part segmentation has practical implications for various applications, including VFX and video editing.

**Weaknesses:**

1. Methodological Clarity: While the overall explanation is detailed, some of the methodology is difficult to follow, especially for readers not familiar with diffusion models and the previous work SLiMe. The process of how tracking integrates into the attention map refinement could be clearer, with more intuitive descriptions of its working mechanism.
1. Dataset & Baselines: The new SMITE-50 dataset is an important contribution, but it appears somewhat small (with 50 videos). Expanding the dataset size or providing an analysis of its generalization for more categories could strengthen the paper’s claims. A comparison to more widespread benchmarks for video segmentation, such as DAVIS or other VOS datasets, would help position SMITE relative to not only a niche dataset but also larger, well-accepted benchmarks in the community.
1. Inference Efficiency: The paper does not deeply explore the computational cost or inference time of the method in practical settings, which would be relevant given the addition of several components (e.g., temporal voting, low-pass filters). Addressing this in the discussion or providing benchmarks on efficiency could help increase understanding of real-world applicability.

**Questions:**

1. How sensitive is the method's performance to the choice of tracking algorithm? Have you experimented with other tracking methods besides CoTracker?
1. The low-pass regularization is mentioned as a key component. Can the authors elaborate on how this regularization term is formulated and how it balances preserving segment structures with smoothing transitions?
1. How does the proposed method handle videos with rapid movements, severe occlusions, or significant lighting changes? Are there strategies to mitigate potential performance degradation in such scenarios?

---

> ### Author Response · Authors · 2024-11-26
> **Official Comment by Authors**
>
> Thank you for appreciating the importance of our work and providing insightful feedback.
>
> **W1: "clarity issue" :** Thanks for your input! For better clarification of our idea, we have now updated the text in lines L267-288 to discuss tracking integrates into the attention map and added lines L154-158 to explain how SLiMe works. We also added minor clarifications about the regularization terms in L344-347.
>
> **W2: "expand dataset or analyze generalization" :** Thanks for your input! While SMITE-50 dataset is planned to be expanded into more classes in the future, it is a challenging dataset and we believe it has its own merits (e.g., number of segmentation parts per video) as compared to the existing video segmentation benchmarks like DAVIS and PumaVOS having more parts per video and many multi-part videos as shown in the table below.
> | Dataset | DAVIS-2016 | PUMAVOS | SMITE-50 |
> |---------|------------|---------|----------|
> | **# of training/testing videos** | 30/20 | 0/24 | 0/50 |
> | **Avg Parts per video** | 1 | 1.08 | 6.39 |
> | **Number of multi-part videos** | 0 | 1 | 50 |
>
> For generalization to other categories, please refer to *"Cross-category generalization"* in Section 7.2 of Appendix. As per request, we have additionally tested our proposed SMITE model on DAVIS dataset in Section 7.3.
>
> **W3: "time analysis" :** We would like to clarify that we do not claim our method is capable of real-time segmentation in the paper. We have now added "Time and Memory Analysis" section in the Appendix of the paper and also the general response.
>
> **Q1: "experiment with other tracking methods" :** In our experiments, we have used the state-of-the-art tracking algorithm CoTracker2 [2]. In addition to this, we have tested on the recently released CoTracker3 [1] and found no significant performance differences. While our method can leverage the benefit of any available tracking algorithm, it will inherit its drawback to some extent. Our low-pass regularization helps partially correct tracker errors since it encourages to respect the semantic segmentation obtained from the diffusion model.
>
> **Q2: "explanation of low-pass regularization" :** Since applying tracking directly (using a voting mechanism with the aid of a tracking method as a post-processing step), may suffer from gradual drift in segmentation, we added low-pass regularization.
> The low-frequency component captures the global structure of segments, helping to prevent segment drift. This regularization ensures consistent boundaries and maintains the overall segmentation structure during tracking (refer to Figure 8 for more clarity). As a result, we have two different energy functions: one pushes the prediction toward removing flicker and achieving more consistency across time (tracking), while the other pushes toward preserving correct semantics and avoiding any possible drift (low-pass regularization).
>
> **Q3: "rapid movements, occlusions, and lighting changes" :** In fact, this is one of the advantages of our method. We have provided several examples to showcase these capabilities. Please see example videos featuring rapid movement (e.g., a running horse in 0:47 to 0:52 of Qualitative Comparison video in the supplementary) and occlusion (e.g., a hand occluding a face 0:15 to 0:21 of Qualitative Comparison video in the supplementary). Our tracking coupled with temporal voting, which accounts for occlusions, is capable of handling such scenarios. Since we use a Diffusion Model with strong semantic knowledge for images and videos under different conditions, our method performs robustly under different lighting conditions. For example, in a camouflage scenario (e.g., Figure 14), although we trained our model on usual turtle examples, our method still performs well in a environment with challenging lighting condition.
>
>
>
> [1] Nikita Karaev, et al. CoTracker3: Simpler and Better Point Tracking by Pseudo-Labelling Real Videos. arXiv,2024.
>
> [2] Nikita Karaev, et al. Cotracker: It is better to track together. arXiv,2023.

---

> ### Comment · Reviewer_rK61 · 2024-11-27
>
> I have read the authors' response and appreciate the clarifications provided. The authors have adequately addressed my concerns regarding the clarity of the methodology, providing a more detailed explanation of the tracking mechanism and its integration with the attention maps. The added information on SLiMe also helps improve overall understanding.
>
> While the dataset size remains a concern, the authors' justification regarding the number of parts and multi-part focus is reasonable. The inclusion of cross-category generalization analysis in the Appendix and the added experiments on the DAVIS dataset further strengthen the paper. The response regarding inference time clarifies the scope of the work.
>
> The authors have also adequately responded to my questions, particularly elaborating on the low-pass regularization and providing examples demonstrating robustness to rapid movements, occlusion, and lighting changes. The rationale for bidirectional tracking is well-explained and justified.
>
> Overall, the revisions have improved the paper and addressed my concerns. I am raising my score.

---

> > ### Author Response · Authors · 2024-11-27
> >
> > Thank you for your thoughtful review and for raising your score.

---

### Official Review · Reviewer_9Vaw · 2024-11-05

**Soundness:** 3
**Presentation:** 3
**Contribution:** 3
**Rating:** 6
**Confidence:** 4

**Summary:**

This paper defines a task of flexible granularity, which aims to segment user-intended granularity temporally using one or a few reference images. To do so, it proposes several techniques to achieve temporal consistent multi granularity segmentation including alternative learning of text embeddings and cross attention module, tracking modules with voting mechanism to be aware of neighbor pixels and low frequency regularizer for spatial consistency. It shows its effectiveness on the proposed dataset, SMITE-50

**Strengths:**

Belows are the strong points that this paper has:

- It is well-written and well-organized, which help reader to understand new task and motivation of the proposed method.

- The proposed techniques including learning generalizable segments using alternative tuning of text embeddings and cross attention, temporal consistency via tracking voting mechanism, and spatial consistency with low-pass filter are in good harmony, which can outperform the other baselines with high margin.

**Weaknesses:**

Belows are points that the reviewer feel concerned about.

- In line #247, the authors mention that learning text embeddings and cross-attention layers is conducted in two phases to provide a better initialization for the next phase. However, this phased or alternating training approach could also increase training complexity. Could the authors provide an ablation study to compare the results of joint training versus alternating training to justify the chosen training setup? Specifically, analysis on training time with final performance of two training strategies can be provided.

- In the supplementary video, specifically at 1:46 in the "Qualitative_Comparison" section, it appears that the proposed method captures a relatively smaller region of the "eye" area compared to the baseline. Is this due to the proposed voting mechanism diluting the effective area by averaging nearby major segments? Would this issue be mitigated by replacing the current hard voting with a soft voting mechanism? Is it doable that quantitative analysis on different voting system can be measured?

- Is the proposed method shape-agnostic within the same class? To demonstrate that the method can segment other unseen images, the authors should provide test samples that differ in shape from the reference object. For example, as similar to what the author showed in "Cars" of Figure 7, if the reference image includes a detailed segmentation map of a "sedan" car, the test samples can be replaced with different shapes of cars, such as "SUVs?

**Questions:**

- Is the proposed method also applicable to "Multi-object flexible granularity", which indicates that reference images annotated with multiple objects (or multiple classes) different granularity can be used as input to the proposed model?

- Could the author provide any failure cases?

---

> ### Author Response · Authors · 2024-11-26
> **Official Comment by Authors**
>
> Thank you for appreciating the importance of our work and providing insightful feedback.
>
> **W1: "ablation for joint vs. two phases":** We would like to clarify that we do not perform two fully separate training stages. Initially, we optimize the text embeddings while the cross-attention components are frozen. Then, we shift the focus to optimizing the cross-attentions of the UNet. During this phase, we continue updating the text embeddings, but at a slower pace (every 10 steps). As such, the initial step serves more as a warm-up phase. Whereas, joint training refers to optimizing the text embeddings and cross attentions together from scratch.
>
> Please refer to the *“Training Ablation: Joint Training vs. Two-Phase Training”* section in Appendix Section 7.2 of the updated paper. Table 7 demonstrates that two-phase training outperforms joint training, achieving 75.14 mIoU compared to 62.17 mIoU, due to better convergence. Additionally, two-phase training leads to a more stable training process with less loss oscillation during training (Figure 9).
>
> **W2: "smaller eye region ; Hard vs. soft voting"** Thank you for pointing this out. We discovered a minor issue during the visualization of this specific category of results, particularly those shown at 1:46 of submitted video. It is common practice [1] to use a small Gaussian blur for improved visualization, but we unintentionally set it to a larger value, which affected small segments such as the 'eyes.' However, please note that this only impacted the visualization; the quantitative results were produced without Gaussian blur. We have now updated the video to exclude the Gaussian blur, and now the eyes are segmented appropriately. In addition to this, we have also provided a comparison image in Supplementary file to show that our model segments out the eye accurately wrt the ground-truth eye segment.
>
> **W3: "shape-agnostic within class, ability to segment unseen samples":** Please refer to the *“Cross-category generalization”* experiment in Section 7.2. As discussed, our method can handle shape variations such as horse to camel or sedan to SUV as illustrated in Figure 9. We have added a discussion in Appendix and added Figure 10 to show this capability.
>
> **Q1: "Use multi-object annotations as input?":** Thanks for your suggestion.  Yes, our model can handle multi-object annotations together. We have now included a discussion on *“Multi-Object Flexible Granularity”* in Section 7.2 and added a new visual result (Figure 11). As shown, SMITE is able to handle multi-object segmentation.
>
> **Q2: "failure case":** Please see the general response above and refer to Section 6 and also Figure 13.
>
> [1] Koichi Namekata, et al. Emerdiff: Emerging pixel-level semantic knowledge in diffusion models. ICLR, 2024.

---

> ### Comment · Reviewer_9Vaw · 2024-11-28
>
> Thanks to the authors for the efforts to resolve my initial concerns and curiosity. The authors specifically clarified the proposed pipeline and presented additional insights to what all the reviewers suggested. I will maintain my recommendation for acceptance.

---

> > ### Author Response · Authors · 2024-11-29
> >
> > Thanks for maintaining your positive score and for your great questions and comments!

---

### Author Response · Authors · 2024-11-26
**General Response for Rebuttal**

We sincerely thank all the reviewers for their valuable feedback and excellent suggestions.

We appreciate their recognition of our paper as well-written ($R_{9Vaw}$), well-structured ($R_{rK61}$), innovative ($R_{rK61}$), novel ($R_{K95K}$), interesting ($R_{K95K}$), and reasonable ($R_{askr}$). Additionally, we are grateful for their acknowledgment of the state-of-the-art performance of our method ($R_{9Vaw}, R_{aCvua}, R_{K95K}$) and the merit of our proposed dataset ($R_{K95K}, R_{askr}$).

We revised our paper to incorporate reviewer feedback (revisions are denoted in blue text). Here, we address the shared concerns and comments raised by the reviewers. Individual reviewer questions are answered under each review. We are happy to respond to any additional questions and welcome further suggestions or updated opinions from the reviewers.

**More ablations on the entire SMITE-50** ($R_{askr}, R_{K95K}$):
We have merged previous Tables 2 and 3 and added a new table (current Table 2) that shows the impact of each component of our method on the entire SMITE-50 dataset. It is evident that our method performs the best when all of our components are being utilized. Please note that low-pass regularization loss without tracking has no impact on the results. Therefore, we did not report the ablation number for that setting.


**Comparison with more baselines including variants of SLiMe ($R_{K95K}$) :** We have updated Table 1 to include more baselines including SLiMe+CoTracker and SLiMe+Inflated UNet. As it is evident, SMITE still outperforms all these other baselines. Please refer to Section 5, Quantitative comparison. We also updated our user study section and Table 4 and we are still ranked highest among the alternatives.

**Time and memory analysis ($R_{aCvu}, R_{9Vaw}, R_{rK61}, R_{askr}$) :** Using 10 images with a batch size of 1 on 15 GB of GPU VRAM, SMITE's training takes 20 minutes. By increasing the batch size and utilizing more GPU VRAM, training time can be reduced to 7 minutes. We train for 300 epochs, but the best accuracy is often reached before 200 epochs, therefore this training time is an upper bound for the samples we showed in the paper. For inference, each frame takes 26 seconds and it requires 60 GB of GPU VRAM to process the entire video. We have added a discussion about time and memory analysis in Appendix.

**Comparison on other datasets including DAVIS ($R_{rK61},R_{aCvu}$) :** To demonstrate SMITE's performance on other datasets compared to the state-of-the-art in video segmentation, we had already included a comparison with PumaVOS in our original submission (Section 7.2). To further compare our method with SOTA for single-object segmentation, we evaluated it on DAVIS-2016. Since DAVIS-2017 (as suggested by aCvu) involves multiple instances, our method cannot be directly applied to DAVIS-2017 without additional modifications. Therefore, our experiments are performed on DAVIS-2016. It is important to note that the strength of our method lies in its ability to perform arbitrary granularity and multi-granular segmentation, while DAVIS-2016 contains only single objects without varying levels of granularity. Therefore, DAVIS-2016 does not fully capture the advantages of our approach. Nonetheless, we discussed this in Appendix; Section 7.3, where we also presented our method's performance in both semi-supervised and zero-shot settings. Despite most alternative methods being specifically trained for video segmentation using datasets like YouTubeVOS or DAVIS, our approach, which requires only a few frames for training, achieves competitive results.

**Limitations ($R_{aCvu},R_{9Vaw}$) :** SMITE has some limitations, which were discussed in Section 6 such as lower performance on smaller objects and also possibly inheriting the limitations of the tracking mechanism. We expanded that discussion in Section 6 and we reiterate it here. Firstly, as it relies on an off-the-shelf tracking technique, its tracking mechanism and voting system are constrained by the performance of this underlying tracking system. Secondly, the generation of WAS maps involves the use of low-resolution cross-attention mechanisms, which may result in reduced accuracy when segmenting fine details or small segments, such as thin lines or minor components. Thirdly, it cannot offer an interactive segmentation since it involves diffusion and optimization processes.

---

### Meta-Review · Area_Chair_4Qk7 · 2024-12-20

**Metareview:**

This paper addresses an interesting problem in the video-object segmentation (VOS) space.

Most VOS methods assume an object query in the first video frame (queries can come in the form of segmentation masks, points, or bounding boxes), and the task is to segment the queried object in the remaining video frames.

Different from that, this paper suggests a "query" that consists of a few images (e.g., three images) that contain an object class of interest (e.g., a horse), with object parts annotated. Given these, the proposed work segment any horse, and its parts, in any video. It can be understood as a few-shot video-object segmentation with image-level supervision.

The method builds upon prior art for a few-shot image segmentation model (SLiMe) that builds upon a text-to-image generation model (Stable Diffusion). On top of that, this paper introduces a video segmentation method based on point tracking and a low-pass regularization term that helps maintain temporal consistency. Empirical validation on a newly introduced dataset, SMITE-50, shows that the proposed network outperforms SLiME baseline (applied per-frame), and several relevant few-shot/zero-shot VOS methods.


Reviewers appreciate that the paper is well-written and organized, find the overall approach innovative, agree that it addresses a new problem in the video object segmentation space, and find that the proposed work has practical applications in the industry (especially VFX and video editing). Overall, there appears to be a consensus that this is a strong paper.

Reviewers also provided constructive feedback. Reviewers note that the paper may not be easy to follow for readers who are not well-versed in the space of generative models or unfamiliar with stable diffusion or prior work this work builds upon (SLiMe).

Reviewers also note that while a new dataset is always a welcome contribution, it is a rather small dataset, and comparisons using canonical VOS benchmarks (DAVIS&related) would strengthen this paper's contributions.

Overall, there is a consensus that this is a good paper (ratings are 8, 8, 6, 6, 6). AC agrees with reviewers; the paper addresses an interesting new VOS task and an interesting model that successfully tackles this problem.

While there was some criticism regarding the empirical validation (reviewers would like to see evaluation beyond the proposed dataset), reviewers agree that the evaluation does support the claims made in the paper. The authors also amended the paper after the rebuttal with additional baselines and results on the DAVIS dataset.

AC suggests revising the paper to make it more accessible to people unfamiliar with SLiMe (the presentation already improved in rebuttal revision, but AC finds more space for streamlining the presentation). By having paper that is self-contained and explanations consolidated and simpler, it will reach a broader audience and have more potential for impact.

The authors responded well to comments regarding dataset size compared to existing VOS datasets and showed that the dataset size is comparable to early VOS datasets. The table provided in the rebuttal does not include more recent and larger datasets (e.g., YT-VOS). Please add this to a revised paper version (it can be in the appendix).

**Additional Comments On Reviewer Discussion:**

While reviews were overall positive, there was a discussion with four reviewers. Reviewers raised a few concerns regarding the scope of the evaluation and asked a few clarifying questions, to which authors responded well. Reviewers acknowledged that and either retained positive initial ratings or upgraded their ratings.

The only exception is reviewer aCvu, which asked for comparisons with more few-shot and zero-shot methods; the authors pointed the reviewer to the table that reports such baseline comparisons and included two additional baselines. The reviewer did not respond, and the review was not updated (initial and final rating was 6).

---

### Decision · Program_Chairs · 2025-01-22

Accept (Poster)